# Geospatial Artificial Intelligence (GeoAI) and Satellite Imagery Fusion for Soil Physical Property Predicting

**Fatemeh Sadat Hosseini [1], Myoung Bae Seo [2,3], Seyed Vahid Razavi-Termeh [2], Abolghasem Sadeghi-Niaraki [2], Mohammad Jamshidi [4] and Soo-Mi Choi [2,\***

[1]  Geoinformation Technology Center of Excellence, Faculty of Geodesy and Geomatics Engineering, K.N. Toosi University of Technology, Tehran 19697, Iran; fatemesadat1476@gmail.com

[2]  Department of Computer Science & Engineering and Convergence Engineering for Intelligent Drone, XR Research Center, Sejong University, Seoul 05006, Republic of Korea; razavi@sejong.ac.kr (S.V.R.-T.); a.sadeghi@sejong.ac.kr (A.S.-N.)

[3]  Future & Smart Constrction Division, Korea Institute of Civil Engineering and Building Technology, Goyang-si 10223, Republic of Korea

[4]  Soil and Water Research Institute (SWRI), Agricultural Research, Education and Extension Organization (AREEO), Karaj 31785-311, Iran; mohammadjamshidi@yahoo.com

*  Correspondence: smchoi@sejong.ac.kr

**Abstract:** This study aims to predict vital soil physical properties, including clay, sand, and silt, which are essential for agricultural management and environmental protection. Precision distribution of soil texture is crucial for effective land resource management and precision agriculture. To achieve this, we propose an innovative approach that combines Geospatial Artificial Intelligence (GeoAI) with the fusion of satellite imagery to predict soil physical properties. We collected 317 soil samples from Iran's Golestan province for dependent data. The independent dataset encompasses 14 parameters from Landsat-8 satellite images, seven topographic parameters from the Shuttle Radar Topography Mission (SRTM) DEM, and two meteorological parameters. Using the Random Forest (RF) algorithm, we conducted feature importance analysis. We employed a Convolutional Neural Network (CNN), RF, and our hybrid CNN-RF model to predict soil properties, comparing their performance with various metrics. This hybrid CNN-RF network combines the strengths of CNN networks and the RF algorithm for improved soil texture prediction. The hybrid CNN-RF model demonstrated superior performance across metrics, excelling in predicting sand (MSE: 0.00003%, RMSE: 0.006%), silt (MSE: 0.00004%, RMSE: 0.006%), and clay (MSE: 0.00005%, RMSE: 0.007%). Moreover, the hybrid model exhibited improved precision in predicting clay ($R^2$: 0.995), sand ($R^2$: 0.992), and silt ($R^2$: 0.987), as indicated by the $R^2$ index. The RF algorithm identified MRVBF, LST, and B7 as the most influential parameters for clay, sand, and silt prediction, respectively, underscoring the significance of remote sensing, topography, and climate. Our integrated GeoAI-satellite imagery approach provides valuable tools for monitoring soil degradation, optimizing agricultural irrigation, and assessing soil quality. This methodology has significant potential to advance precision agriculture and land resource management practices.

**Keywords:** machine learning; deep learning soil texture; satellite imagery; geospatial analysis; land resource management

## 1. Introduction

Soil is a crucial component of climate and ecosystem regulation and a fundamental factor in producing 97% of human food [1]. Soil also significantly impacts agricultural productivity, watershed protection, the environment, and wildlife [2]. Soil texture is critical in soil erosion, water transfer, quality control, and productivity. The particle size classification of soil texture includes sand (2–0.05 mm), silt (0.05–0.002 mm), and clay (<0.002 mm) [3]. Among the significant challenges facing soil are soil erosion and rainfall

erosion at different scales, which can alter soil properties, and particularly its texture [4]. Therefore, spatial prediction of soil properties is crucial for evaluating soil quality that human use affects [2].

Remote sensing (RS) data are a globally available and abundant source of information that is highly valuable in agriculture. Advances in RS technology have significantly improved data processing at large spatial and temporal scales [4]. Aerial images and digital image processing were previously used to monitor agricultural land. However, RS now allows for reducing collected field data while improving estimates' accuracy and efficiency [5]. In conjunction with Geographic Information Systems (GIS), RS can increase the efficiency of collection, storage, analysis, and modeling in terms of cost, time, and human resources [6].

Additionally, GIS provides various tools for combining spatial information and environmental parameters to aid spatial prediction. It is also an effective analysis tool for mapping, data management, and spatial analysis [7]. RS and GIS data can be used as predictive variables for the spatial modeling of a phenomenon [8]. Recent years have seen significant advancements in using spatial information systems and RS tools or features in predicting soil properties [9].

Various statistical and geostatistical methods, such as kriging [10], multiple stepwise regression [11], partial least squares regression [12], and cokriging [11], have been previously used to predict the spatial distribution of soil texture. However, these methods heavily rely on statistical assumptions and become computationally intensive with increasing data size [13]. Machine Learning (ML) algorithms have been applied to predict soil texture properties to overcome these limitations. ML algorithms, such as regression trees [14], Boosted Regression Trees (BRT) [15], Random Forest (RF) [16], and Support Vector Machine (SVM) [15], have demonstrated their capability in mapping soil texture properties. ML algorithms offer significant advantages in managing high-dimensional and multi-variable data by discovering and identifying implicit relationships [17]. However, despite their benefits, these algorithms are prone to problems such as providing only locally optimal solutions, decreased performance when training time is extended, and difficulty finding the optimal learning rate [18].

While some ML algorithms may exhibit saturation in performance as the data volume increases, the relationship between data volume and algorithm performance is influenced by various factors, including the heterogeneity and relevance of the data. In cases where data are diverse and contain valuable information across different scales and contexts, increasing data volume can enhance model performance. However, it is essential to carefully curate and preprocess the data to ensure that the additional volume contributes meaningfully to model training and generalization. Moreover, ML algorithms cannot detect irrelevant and redundant information, which negatively impacts their performance [17]. While ML can handle complex data, excessive hidden layers can lead to issues such as overfitting and vanishing gradients [19,20]. DL, with its strong predictive accuracy, outperforms ML in spatial prediction. To tackle intricate soil challenges, sophisticated algorithms such as Convolutional Neural Network (CNN), rooted in DL, are used to boost accuracy and reduce uncertainty [21,22]. Additionally, DL networks offer automatic information extraction capabilities not present in ML models [23]. Overall, DL addresses the shortcomings of ML by providing enhanced performance, automatic feature extraction, and improved scalability. Various researchers have utilized DL models to address soil science problems such as predicting soil texture [24,25] and soil salinity [26] using the CNN algorithm and predicting soil moisture using the LSTM algorithm [27]. While DL models have several advantages, they are also associated with drawbacks such as computational complexity [28] and overfitting [29]. Researchers have proposed combining DL models with ML algorithms to overcome these limitations. In such combined networks, the hierarchical nature of DL models enables them to automatically extract essential features from raw data, while ML algorithms process regression operations more efficiently than DL models, thus solving the disadvantages of each [30,31]. Despite the pros and cons associated with ML

and DL models, the amalgamation of these two approaches has been widely employed across various research domains. For instance, CNNs and RF combinations have been applied in early earthquake warning systems [32] and poverty estimation using satellite imagery [33].

Additionally, ML algorithms have been integrated with DL neural networks to estimate flood potential [34], while CNNs have been combined with support vector machine, RF, and logical regression to evaluate landslide susceptibility [35], leading to improved performance and accuracy of results. Therefore, in this research, a combination of two algorithms, RF and CNN, has been utilized to enhance the accuracy in the spatial prediction of soil texture properties. In addition to overcoming overfitting, the RF algorithm exhibits acceptable accuracy compared to other ML algorithms in spatial modeling [36]. On the other hand, the CNN algorithm can automatically extract various features, particularly spatial features, by processing information through convolution layers [37,38].

## 2. Materials and Methods

### 2.1. Study Area

The study area in the Golestan province of Iran spans from latitude 36°56′ to 37°35′ and from longitude 54°58′ to 55°42′ (Figure 1). Most of the region, including its central and northwestern parts, is dedicated to wheat cultivation, while the southern and northeastern areas are primarily used for grazing. The highest and lowest elevations in the area are 1722 and 0 meters above sea level, respectively. The average annual rainfall and air temperature in the study area are 456 mm and 21 °C, respectively. Our study aims to predict soil physical properties using a fusion of methods and data sources, focusing on the unique challenges the study area poses. The target spatial resolution for our predictions is 30 × 30 m, which reflects the scale at which we aim to generate predictive maps.

### 2.2. Soil Samples

This study's soil sample data consist of 317 samples (0–30 cm) collected by the Iran Water and Soil Research Institute, including the three properties of clay, sand, and silt. The sampling was conducted using the grid sampling method, with each grid covering an area of 1 km² and the precise coordinates of the soil samples determined using the Global Positioning System (GPS). In total, 317 soil samples were distributed across various landcover classes (Figure 2). Specifically, 73% of the samples belonged to agricultural land, 13% to range land, 9% to uncovered plain, 3% to residential areas, 1% to forest, and 1% to water bodies. Out of all the soil samples, approximately 75% were situated at altitudes below 200 m, while the remaining 25% were located at altitudes above 200 m.

The hydrometer method [39] was used to analyze soil texture properties, including sand, silt, and clay. Table 1 presents the soil texture properties' minimum, maximum, mean, and standard deviation values.

**Table 1.** Statistical summary of soil texture.

| Soil Texture | Clay (%) | Silt (%) | Sand (%) |
|---|---|---|---|
| Minimum | 0 | 0 | 0 |
| Maximum | 44 | 80 | 58 |
| Mean | 22.322 | 64.457 | 12.867 |
| Standard deviation | 6.920 | 9.249 | 9.115 |

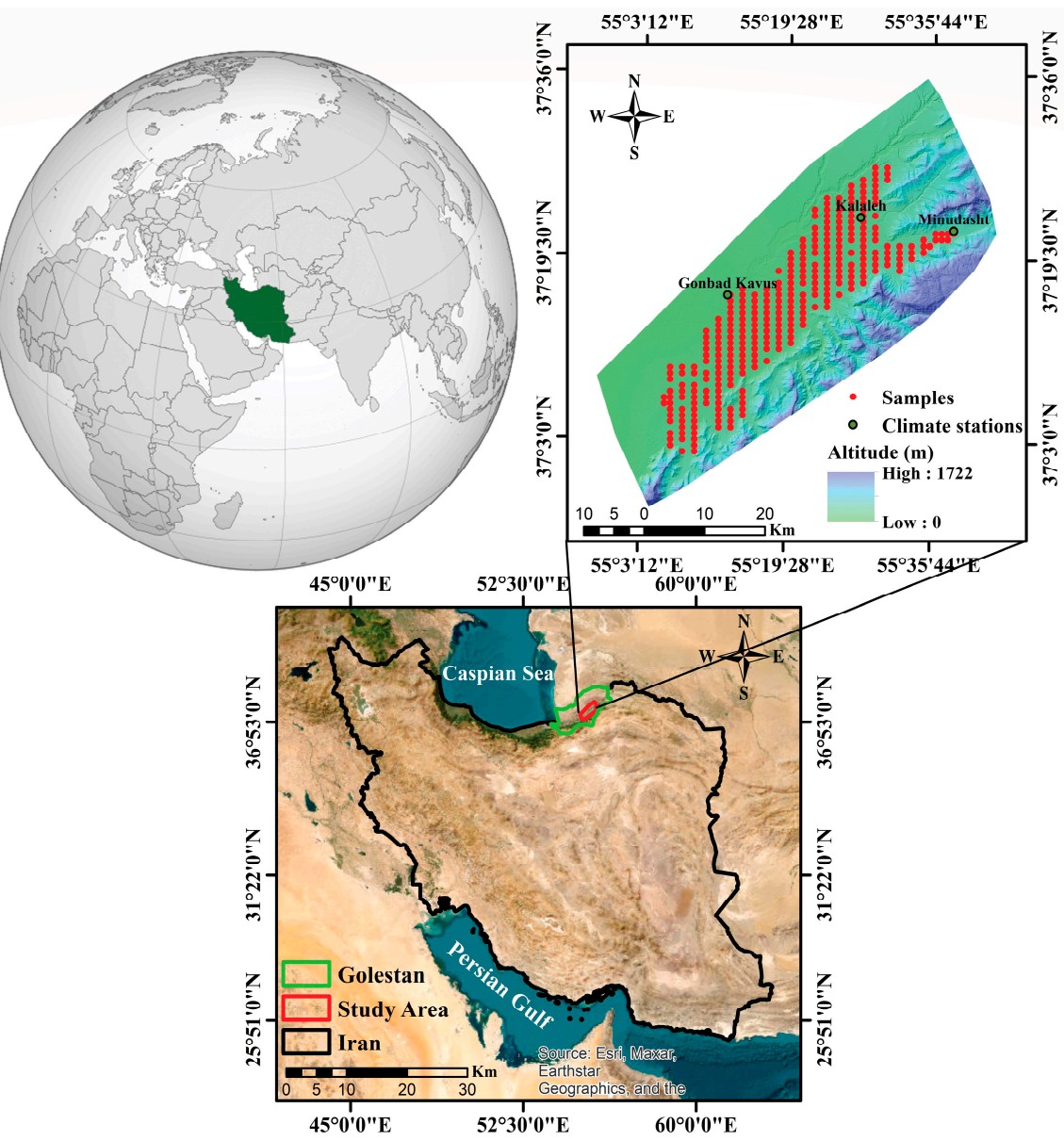

**Figure 1.** Study area location, distribution of soil samples, and meteorological stations.

Table 2 presents a summary of the statistical data for each type of soil texture after removing any outliers from the dataset using Theil–Sen regression. The original dataset consisting of 317 soil samples was reduced to 179, 144, and 155 samples for sand, silt, and clay, respectively.

**Table 2.** Statistical summary of soil texture after outlier removal.

| Soil Texture | Clay (%) | Silt (%) | Sand (%) |
|---|---|---|---|
| Minimum | 12 | 50 | 4 |
| Maximum | 36 | 76 | 26 |
| Mean | 22.182 | 65.74 | 11.056 |
| Standard deviation | 4.199 | 4.567 | 3.705 |

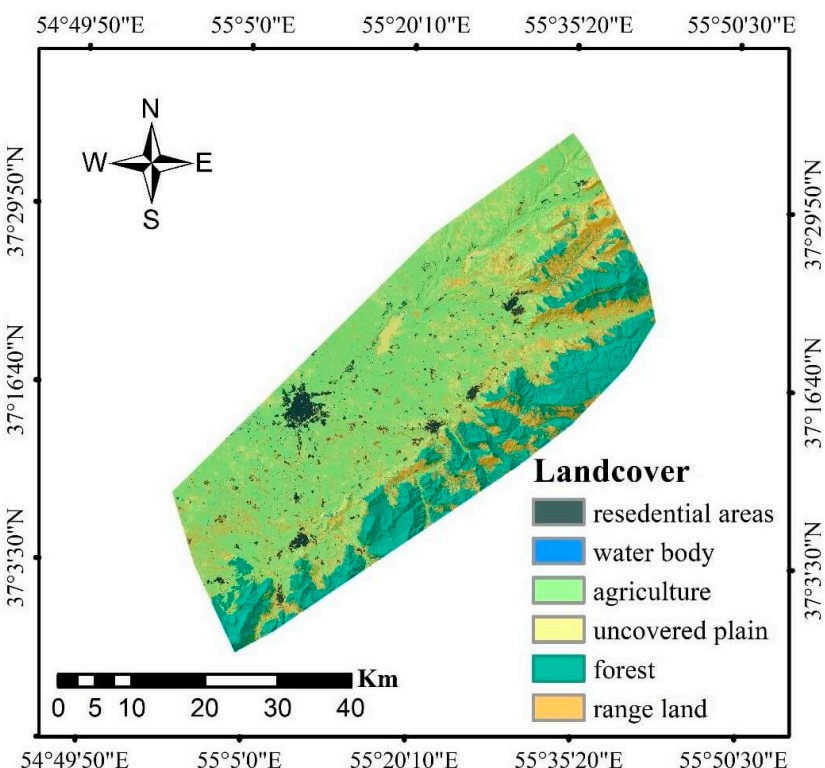

**Figure 2.** Landcover classes in the study area.

*2.3. Environmental Parameters*

Based on previous studies [22,40,41], expert opinions, and the specific conditions of the studied area, three groups of environmental parameters were used. These included RS variables such as Band 1 (B1) to Band 5 (B5) and Band 7 (B7) of Landsat-8, Brightness Index (BI), Coloration Index (CI), Clay Index (CLI), Enhanced Vegetation Index (EVI), Land Surface Temperature (LST), Hue Index (HI), Normalized Difference Vegetation Index (NDVI), Redness Index (RI), and Saturation Index (SI). Climate variables such as air temperature and rainfall were also included, along with topographic variables including aspect, elevation, slope, duration radiation (DR), the Multi-Resolution index of Valley Bottom Flatness (MRVBF), the Multi-Resolution Ridgetop Flatness index (MRRTF), and the Topographic Wetness Index (TWI). In this study, dependent parameters represent soil texture properties, which are treated as target variables, and independent parameters encompass various environmental parameters (Table 3).

**Table 3.** The parameters that impact the properties of soil texture.

| Soil Texture | Effective Parameters | Number of Parameters |
|---|---|---|
| Clay | NDVI, Elevation, B7, B5, B1, B2, B3, B4, MRRTF, MRVBF, Rainfall, SI, CI, LST, Temp, Aspect, RI, TWI | 18 |
| Silt | NDVI, Elevation, B7, B5, B3, B4, MRRTF, MRVBF, SI, BI, CLI, CI, Slope, EVI, DR, Aspect, RI, TWI | 18 |
| Sand | NDVI, Elevation, B7, B5, B1, B2, B3, B4, Rainfall, SI, BI, CLI, MRRTF, MRVBF, CI, Slope, LST, DR | 18 |

2.3.1. RS Parameters

For this study, 14 RS parameters were extracted from Landsat 8 satellite images, as listed in Table 4. The RS images utilized were collected between 1 January and 30 December

2020. The image locations correspond to path 162, row 34, path 162, row 35, and path 163, row 34 of the Landsat global reference system. The Landsat 8 OLI sensor images were radiometrically and geometrically corrected in Google Earth and projected to WGS84-Zone 40 N.

**Table 4.** RS parameters.

| Covariate Name | Definition | Reference |
|---|---|---|
| Coastal aerosol (B1) | 0.43–0.45 μm | |
| Blue (B2) | 0.45–0.51 μm | |
| Green (B3) | 0.53–0.59 μm | |
| Red (B4) | 0.64–0.67 μm | [41] |
| Near-infrared (B5) | 0.85–0.88 μm | |
| Short-wave infrared-2 (B7) | 2.11–2.29 μm | |
| Brightness Index (BI) | $\left(B3^2 + B4^2\right)^{0.5}$ | [42,43] |
| Clay Index (CLI) | $B6/B7$ | [22] |
| Coloration Index (CI) | $(B4 - B3)/(B4 + B3)$ | [42,44] |
| Enhanced Vegetation Index (EVI) | $2.5 \times \left(\frac{B5-B4}{B5+(6\times B4)-(7.5\times B2)+1}\right)$ | [45] |
| Land Surface Temperature (LST) | | |
| Normalized Difference Vegetation Index (NDVI) | $(B5 - B4)/(B5 + B4)$ | [46] |
| Redness Index (RI) | $\left(B4^2\right)/\left(B2 \times \left(B3^3\right)\right)$ | [44] |
| Saturation Index (SI) | $(B4 - B2)/(B4 + B2)$ | [47] |

2.3.2. Topographic Parameters

The topographic parameters used in this study were extracted from the Shuttle Radar Topography Mission (SRTM) digital terrain model, with a spatial resolution of $30 \times 30$ m, using the Google Earth Engine system and ArcGIS 10.8 and SAGA 8.2.1 software. These parameters included aspect, elevation, slope, Duration Radiation (DR), Multi-Resolution index of Valley Bottom Flatness (MRVBF), Multi-Resolution Ridgetop Flatness index (MR-RTF), and Topographic Wetness Index (TWI). TWI was calculated using Equation (1),

$$\text{TWI} = \ln \frac{A_s}{\tan \beta} \tag{1}$$

where $A_s$ is the catchment area index and $\beta$ is the slope angle [48].

2.3.3. Climatic Parameters

The climatic parameters used in this study were obtained from the annual average (2014–2020) data of 10 Meteorological stations in Golestan province, as shown in Figure 1. Various interpolation methods were applied to the data using ArcGIS 10.8 software. The local polynomial method was the most accurate for generating maps of air temperature and rainfall, based on the RMSE index.

*2.4. Prediction Models*

2.4.1. RF Algorithm

The RF algorithm, developed by Breiman, is an ensemble learning technique that combines the prediction results of multiple decision trees to achieve higher accuracy [49]. This algorithm has been widely used in various fields and has shown excellent performance in solving classification, regression, and unsupervised learning problems [50]. In an RF, a set of tree predictors $h(x; \theta_k)$, $k = 1, \ldots, K$ is used, where $x$ represents the input vector of observations (variables) and $\theta_k$ are independent and identically distributed random

vectors [51]. Each $\theta_k$, which replaces the original data set, is fitted into a regression tree. A small set of input variables is randomly considered for each node in each tree. The tree division criterion is based on selecting the input variable with the lowest Gini index [52]. Finally, the output of the RF prediction in regression problems is the unweighted average of the entire set of decision trees (Equation (2)) [53].

$$h(x) = \left(\frac{1}{k}\right) \sum_{k=1}^{K} h(x; \theta_k) \qquad (2)$$

The overall flowchart of the RF is shown in Figure 3.

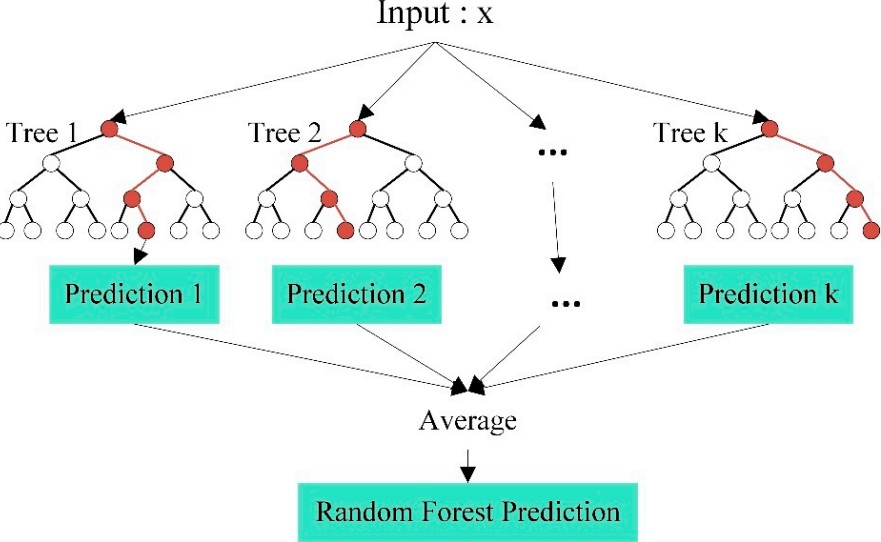

**Figure 3.** Architecture of the RF algorithm.

### 2.4.2. CNN

A CNN is an architecture for DL inspired by living organisms' visual perception mechanism [54]. It consists of several layers: convolution, maximum pooling, dropout, concatenate, and fully connected [55]. The convolution layer contains several kernels that calculate different features from the input data [54]. The top pooling layer sends the maximum number of features of each region as input to the next layer, reducing the dimensionality of the matrix and avoiding overfitting [56]. Dropout is another way to prevent overfitting [57]. Equation (3) calculates the output $C_j$ of the convolution layer, where $x_i$ is the $i$th feature of the input vector of the CNN network, $W_{ij}$ is the weight between $x_i$ and the $j$th kernel of the convolution layer with bias $b$, and $k$ and $n$ are the number of kernels and the number of features of the input vector to the convolution layer, respectively [58]. The activation function $f$ can be sigmoid, tanh, or ReLU, among others.

$$C_j = f\left(b + \sum_{i}^{n} \text{conv1D}(W_{ij}, x_i)\right), j = 1, 2, \ldots, k \qquad (3)$$

Figure 4 depicts the CNN architecture.

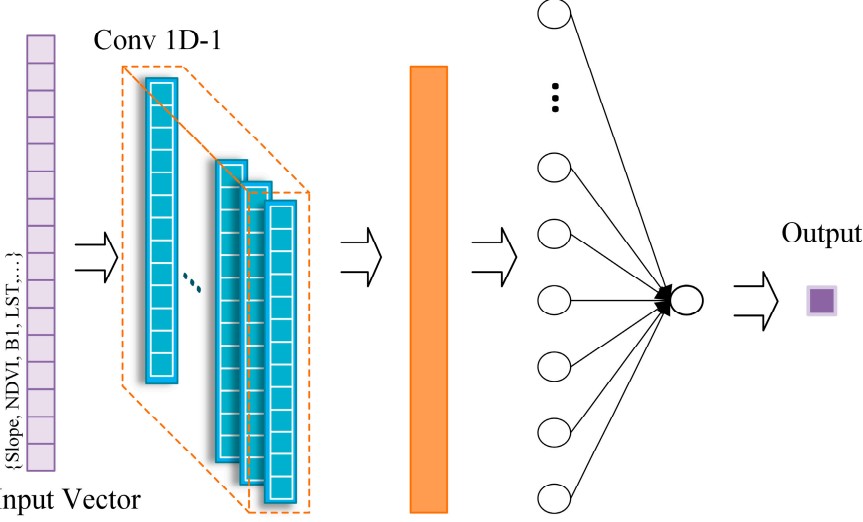

**Figure 4.** Architecture of the CNN model.

### 2.4.3. CNN-RF

In this study, a hybridized network of DL and ML is used to leverage the capabilities of both CNN networks and the RF algorithm to achieve higher performance in the spatial prediction of soil texture and overcome the limitations of these stand-alone models. The hybrid CNN-RF network architecture is shown in Figure 5. The input matrix assumes an $m \times n$ structure, where m signifies the quantity of soil samples and $n$ represents the number of parameters influencing each soil texture property. The input information is first processed through the hidden layers of the CNN model, which extracts the relevant features including the spatial patterns and contextual information from the input dataset [59]. These features are then fed into the RF algorithm for regression analysis. Finally, the output layer returns the predicted value.

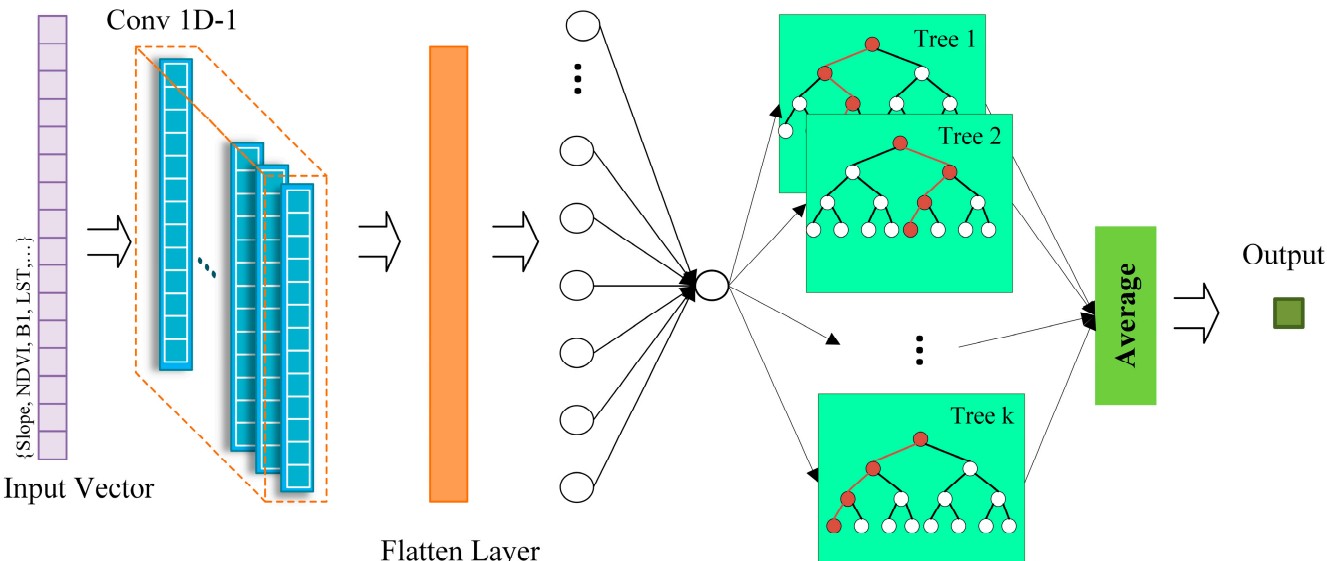

**Figure 5.** Architecture of the CNN-RF algorithm.

### 2.5. Models Evaluation

The efficiency of the model is evaluated using three metrics: Root Mean Squared Error (RMSE), Mean Squared Error (MSE), and coefficient of determination ($R^2$) (Equations (4)–(6)). Lower MSE and RMSE values indicate a higher modeling accuracy. $R^2$ illustrates the goodness of fit between the data and the regression model. The value of $R^2$ ranges from 0 to 1, with values closer to 1 indicating better model performance [60,61].

$$MSE = \frac{1}{n} \sum_{i=1}^{n} (y_i - \hat{y}_i)^2 \tag{4}$$

$$RMSE = \sqrt{\frac{1}{n} \sum_{i=1}^{n} (y_i - \hat{y}_i)^2} \tag{5}$$

$$R^2 = 1 - \frac{MSE}{\frac{1}{n} \sum_{i=1}^{n} (y_i - \overline{y})^2} \tag{6}$$

In these equations, $y_i$ represents the measured value, $\hat{y}_i$ represents the predicted value, $\overline{y}$ represents the mean of the actual values, and $n$ is the number of observations. Another effective way to display the relationship between statistical indicators and to visualize the difference in model performance in predicting soil properties is to use a Taylor diagram [62]. Taylor diagrams show the degree of agreement between predicted and observed values regarding correlation and the standard deviation error. Additionally, a box plot is used to compare the minimum and maximum values of the range, the upper and lower quartiles, and the median of the predicted values and the actual data. This set of values provides a concise summary of the distribution of the dataset [63].

### 2.6. K-Fold Cross-Validation

Cross-validation is a technique used to assess the performance of a machine and deep learning models in a robust and unbiased manner [64]. In 10-fold cross-validation, the dataset is divided into 10 folds of approximately equal size. The dataset is randomly divided into 10 subsets, each containing an equal number of samples. This ensures that the distribution of data across the folds is representative of the entire dataset [64]. The cross-validation process is then performed iteratively, with each fold being used as the testing set while the remaining nine folds are used for training the model. The performance metrics are calculated for each iteration based on the model's predictions.

### 2.7. Workflow for Soil Texture Prediction

The workflow for spatial prediction of soil texture properties is illustrated in Figure 6. The first step involves creating a spatial database using parameters extracted from satellite images and data collected from the study area. In the second step, the extracted parameters are used as independent data to determine feature importance through the RF algorithm. In the third step, soil texture properties are modeled using the RF, CNN, and CNN-RF algorithms. In the fourth step, prediction maps of soil texture properties are generated using the models. Finally, the results are evaluated using five metrics: MSE, RMSE, $R^2$, box plot, and Taylor diagram.

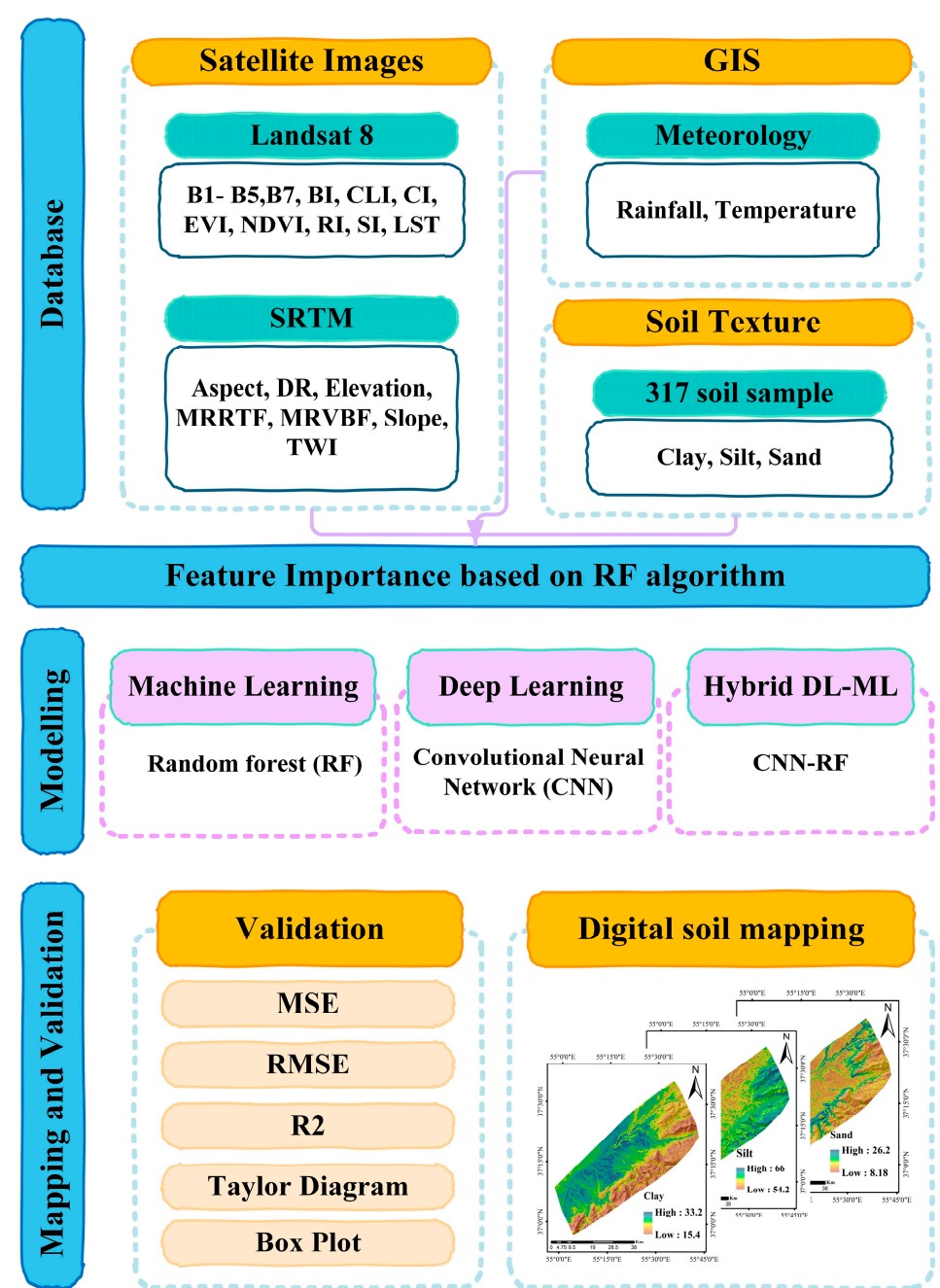

**Figure 6.** Research flowchart.

## 3. Results

### 3.1. Correlation Analysis

In this study, the Pearson correlation coefficient was used to investigate the relationship between soil texture and environmental parameters (Figure 7). According to Figure 7, The correlation coefficient of 0.2 between MRVBF and clay suggests a comparatively stronger relationship between these variables compared to other parameters, while the correlation coefficient of −0.26 between B7 and clay indicates that their relationship in the opposite direction is also relatively stronger than that of other parameters. By contrast, the association between clay and RI, as well as clay and aspect, was considered weak, with absolute correlation coefficients of 0.021 and −0.034, respectively. Based on Figure 7, the correlation coefficients of −0.18 between sand and LST, and 0.12 between sand and B5, demonstrate a comparatively stronger association compared to other parameters. Among all the parameters, MRVBF and CI exhibited the weakest correlation with sand. The

Pearson correlation coefficient matrix in Figure 7 shows that the correlation coefficient of 0.2 between elevation and silt indicates a relatively stronger positive relationship compared to other parameters. Additionally, the correlation coefficient of −0.17 between NDVI and silt suggests a relatively stronger negative relationship. However, the associations between silt and B5, as well as silt and RI, were considered weak, with correlation coefficients of 0.013 and −0.033, respectively.

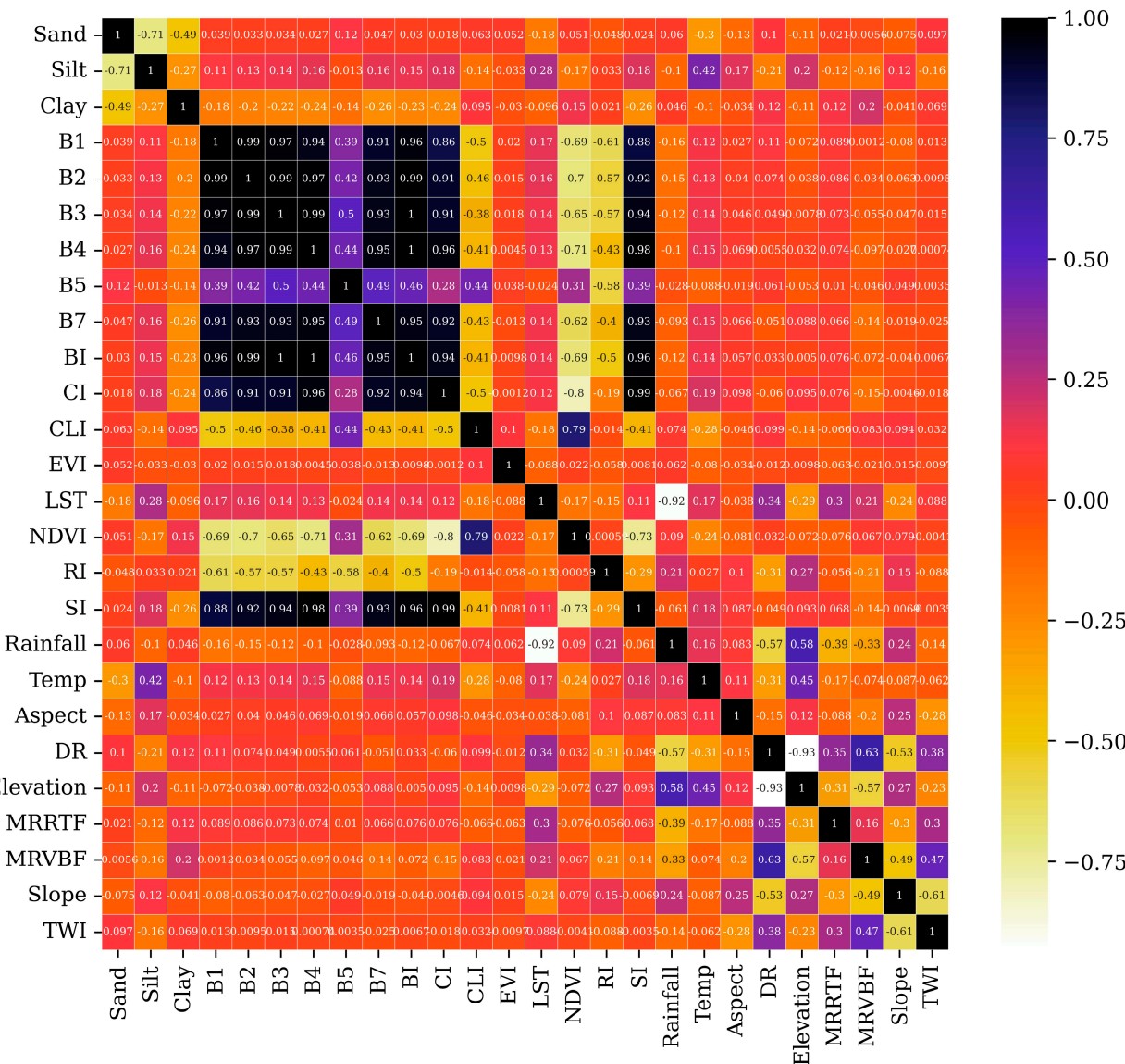

**Figure 7.** The correlation coefficients between environmental parameters and soil texture.

### 3.2. Feature Importance

An RF algorithm was utilized to determine features importance in the modeling process. The importance of parameters is demonstrated in Figure 8. The results indicate that B7 (0.123), CI (0.089), and TWI (0.084) are among the parameters that show a higher association with silt content (Figure 8a). In the case of Sand, LST (0.164), B5 (0.089), and elevation (0.084) exhibit relatively higher importance (Figure 8c). Similarly, MRVBF (0.119), B7 (0.140), and TWI (0.096) are identified as significant factors influencing soil clay (Figure 8e).

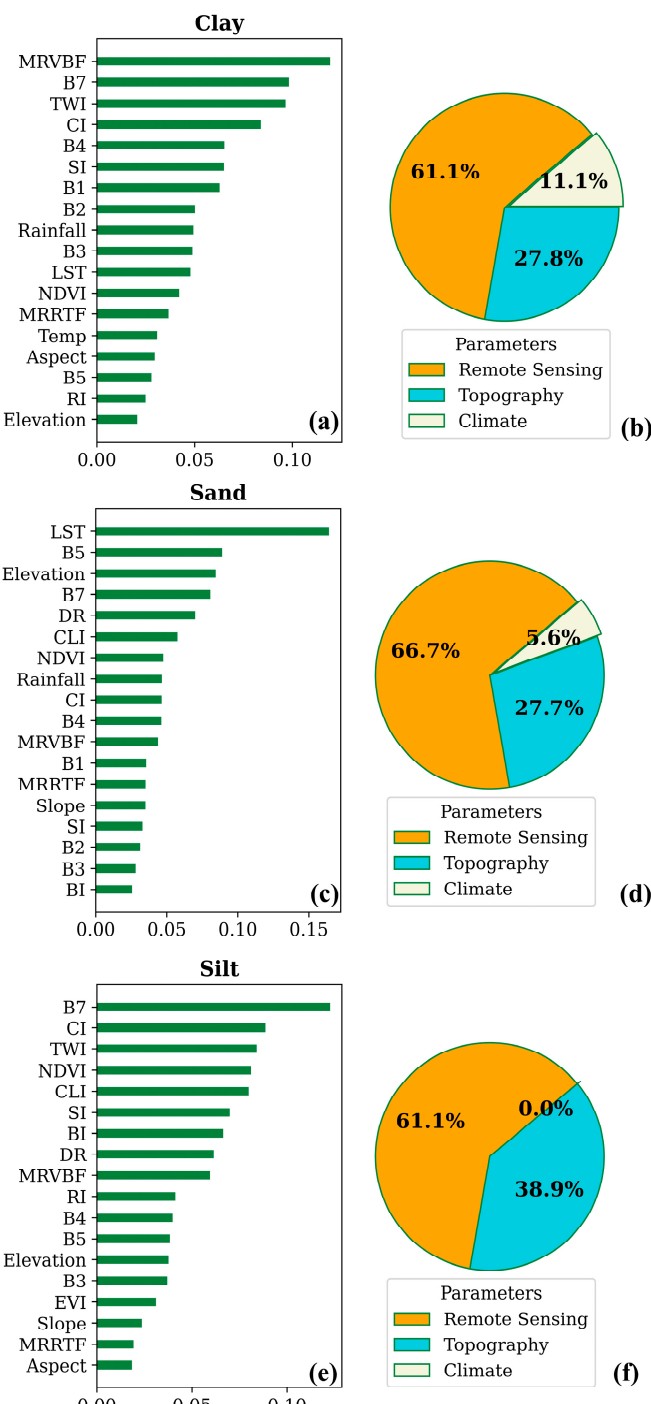

**Figure 8.** Feature importance based on the RF algorithm for soil texture: (**a**) clay, (**c**) sand, and (**e**) silt; and the portion of each environmental category in the input data: (**b**) clay, (**d**) sand, and (**f**) silt.

According to Figure 8, the model input parameters for clay and sand are mainly influenced by RS, topography, and climatic parameters. Among these parameters, RS parameters significantly impact soil texture more than topography and climatic parameters. However, as illustrated in Figure 8f, climatic parameters do not play a role in determining soil silt. Among the climatic parameters, rainfall, while among the RS parameters, NDVI and B7 have the most significant impact on soil texture. In addition, among the topographic parameters, TWI, MRVBF, and MRRTF have the most significant effect on soil texture properties.

### 3.3. Model Development

Using the Python programming language for spatial modeling, the ML and DL models were employed in the Google colab environment (colab.research.google.com, 20 March 2020). The computer for developing models and processing information had an Intel Core with i7 CPU @2.80 GHz and 16 GB of RAM. Various libraries such as Keras, TensorFlow, Numpy, CSV, Scikit-learn, and Matplotlib were utilized for implementing models and generating graphs. The data pre-processing step involved normalization, followed by cross-validation and determination of hyperparameters using the GridSearch method. Equation (7) was employed for normalization, with X denoting the value of each feature. The optimized hyperparameters and layers for the models are listed in Table 5. Data were split using 10-fold cross-validation. The data were split into 10 equal parts, and then one fold was used for the validation set and nine remaining folds were used for the training set. For each fold, the models were trained using the training set and evaluated by the testing set.

**Table 5.** The optimized hyperparameters and layers for each model. "✓" signifies the inclusion of specific layers in the model.

| | | | Filter/ Number of Trees | Filter Size | Activation Function | CNN | RF | CNN-RF |
|---|---|---|---|---|---|---|---|---|
| Layers | L1 | Convolutional | 32 | 3 | ReLU | ✓ | - | ✓ |
| | L2 | Flatten | - | - | - | ✓ | - | ✓ |
| | L3 | Fully connected | 64 | 2 | ReLU | ✓ | - | ✓ |
| | L4 | Fully connected | 1 | - | - | ✓ | - | ✓ |
| | L5 | RF | 100 | - | - | - | ✓ | ✓ |
| Other parameters | | Batch_size | - | - | - | 10 | - | 10 |
| | | Epochs | - | - | - | 20 | - | 20 |
| | | Optimizer | - | - | - | Adam | - | Adam |
| | | Loss | - | - | - | MSE | - | MSE |
| | | min_samples_split | - | - | - | - | 2 | 2 |
| | | max_features | - | - | - | - | 'auto' | 'auto' |
| | | max_depth | - | - | - | - | 'None' | 'None' |
| | | bootstrap | - | - | - | - | 'True' | 'True' |

For modeling, the CNN, RF, and CNN-RF models were used. The input matrix for each model was an $m \times n$ matrix, where $m$ represents the number of soil samples and $n$ indicates the number of parameters affecting each soil texture property.

$$X_{new} = (X_i - Min(X))/(Max(X) - Min(X)) \tag{7}$$

### 3.4. Comparison of Prediction Models

To spatially model soil texture, a combination of the CNN DL model and the RF ML algorithm was utilized. To evaluate the performance of these models, three evaluation metrics, namely MSE, RMSE, and $R^2$, were employed, and the evaluation results are presented in Table 6. The results indicate that for clay, the CNN, RF, and CNN-RF algorithms yielded MSE values of 0.00016%$^2$, 0.00079%$^2$, and 0.00005%$^2$, RMSE values of 0.013%, 0.028%, and 0.007%, and $R^2$ values of 0.981, 0.910, and 0.995 in the training phase, and MSE values of 0.00038%$^2$, 0.00407%$^2$, and 0.00010%$^2$, RMSE values of 0.019%, 0.064%, 0.010%, and $R^2$ values of 0.966, 0.636, 0.982 in the testing phase. Regarding sand, the CNN model produced MSE values of 0.00029%$^2$ and 0.00046%$^2$, RMSE values of 0.017% and 0.022%, and $R^2$ values of 0.928 and 0.908 in the training and testing phases, respectively. Additionally, for this property, the RF algorithm generated MSE values of 0.00034%$^2$ and

0.00135%$^2$, RMSE values of 0.018% and 0.037%, and $R^2$ values of 0.917 and 0.683, while the combined CNN-RF model produced MSE values of 0.00003%$^2$ and 0.00007%$^2$, RMSE values of 0.006% and 0.008%, and $R^2$ values of 0.992 and 0.976 in the training and testing phases, respectively. Furthermore, for silt, the CNN model yielded MSE, RMSE, and $R^2$ values of 0.00024%$^2$, 0.016%, and 0.920, respectively, during the training phase, and 0.00040%$^2$, 0.020%, and 0.913, respectively, during the testing phase. Moreover, the RF algorithm generated MSE values of 0.00022%$^2$ and 0.00060%$^2$, RMSE values of 0.00060% and 0.024%, and $R^2$ values of 0.935 and 0.676 for this property during the testing and training phases, respectively. In comparison, the combined CNN-RF model produced MSE, RMSE, and $R^2$ values of 0.00004%$^2$, 0.006, and 0.987 during the training phase and 0.00009%$^2$, 0.010%, and 0.980 during the testing phase, respectively.

**Table 6.** Evaluation results.

| Properties | Models | Train | | | Test | | | Runtime |
| | | MSE (%$^2$) | RMSE (%) | $R^2$ | MSE (%$^2$) | RMSE (%) | $R^2$ | (s) |
| --- | --- | --- | --- | --- | --- | --- | --- | --- |
| | CNN | 0.00016 | 0.013 | 0.981 | 0.00038 | 0.019 | 0.966 | 2.67 |
| Clay | RF | 0.00079 | 0.028 | 0.910 | 0.00407 | 0.064 | 0.636 | 0.23 |
| | CNN-RF | 0.00005 | 0.007 | 0.995 | 0.00010 | 0.010 | 0.982 | 0.21 |
| | CNN | 0.00029 | 0.017 | 0.928 | 0.00046 | 0.022 | 0.908 | 1.36 |
| Sand | RF | 0.00034 | 0.018 | 0.917 | 0.00135 | 0.037 | 0.683 | 0.44 |
| | CNN-RF | 0.00003 | 0.006 | 0.992 | 0.00007 | 0.008 | 0.976 | 0.29 |
| | CNN | 0.00024 | 0.016 | 0.920 | 0.00040 | 0.020 | 0.913 | 2.73 |
| Silt | RF | 0.00022 | 0.015 | 0.935 | 0.00060 | 0.024 | 0.676 | 0.196 |
| | CNN-RF | 0.00004 | 0.006 | 0.987 | 0.00009 | 0.010 | 0.980 | 0.215 |

The runtime analysis of the three models (Table 6) revealed varying performance when fitting the dataset. Specifically, the CNN model for clay and sand exhibited the longest runtime, followed by RF and CNN-RF. Conversely, when fitting the silt data, the runtime of CNN-RF was found to be longer compared to RF, with CNN once again exhibiting the longest runtime among all soil texture models.

Overall, the results indicate that the hybrid CNN-RF algorithm performs better than the other models in both the testing and training phases for all soil texture properties. After the hybrid CNN-RF algorithm, the CNN model is more accurate than the RF algorithm. Based on the MSE evaluation metric, the sand, silt, and clay properties of soil texture are the most accurate.

The prediction error plots for the testing and training phases are presented in Appendix A. Across all three soil texture properties, the CNN-RF model exhibits lower error rates or differences between the observed and predicted values than the stand-alone models. The RF and CNN models demonstrate a better fit between the actual and predicted value plots.

Figure 9 displays box plots that compare the values predicted by all three prediction models, namely CNN, RF, and CNN-RF, with the actual soil sample values in terms of statistics and data distribution. The lines outside the boxes extend up to 1.5 times the interquartile range to identify any outliers (hollow circles) that lie beyond this range [63]. The median is depicted using a yellow line in the center of the box. As shown in Figure 9, the box plot of values predicted by the CNN-RF model for all three soil texture properties is more similar to the box plot of the observed values. The distribution of actual values of all data is nearly symmetrical, and the predicted values for all three prediction models are also symmetrically distributed. Furthermore, the CNN-RF and CNN models are better at detecting and predicting outlier data than the RF model.

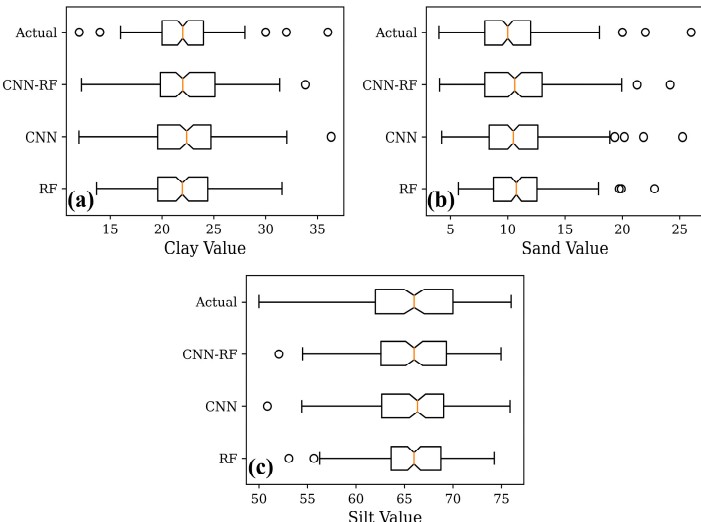

**Figure 9.** Box plots for comparison of the hybrid CNN-RF, RF, and CNN models for soil properties: (**a**) clay, (**b**) sand, (**c**) silt.

Taylor diagrams were employed to assess the accuracy of the CNN, RF, and CNN-RF models, as depicted in Figure 10. A smaller distance from the purple reference point in Taylor diagrams indicates a higher model accuracy [62]. Consequently, a model's accuracy is determined based on the distance of the corresponding point from the purple reference point. According to the Taylor diagrams in Figure 10, the hybrid CNN-RF model exhibits the most accurate prediction for all three soil texture properties followed by the CNN and RF models, sequentially.

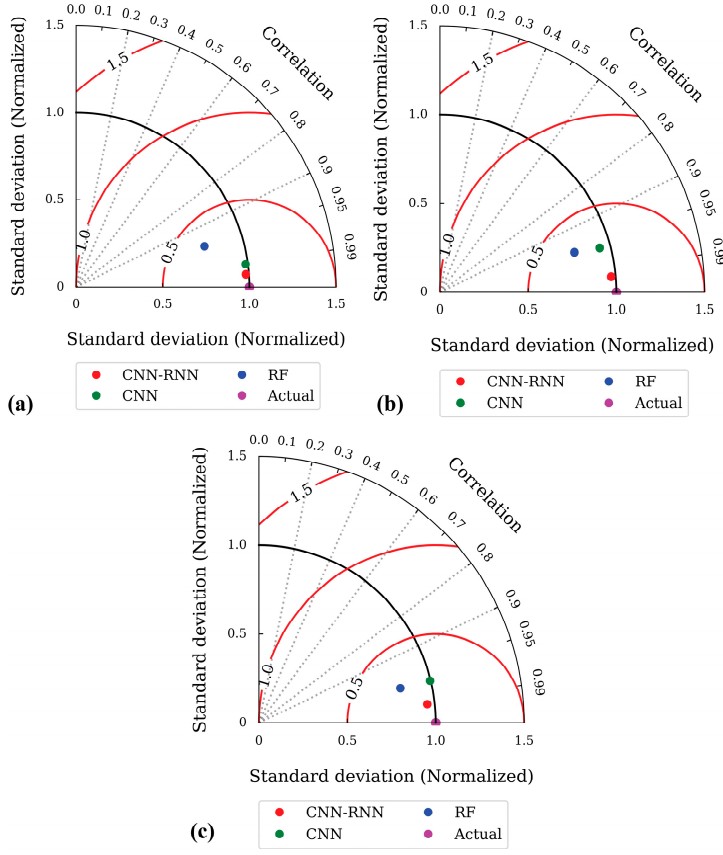

**Figure 10.** Taylor diagrams for comparison of model performance: (**a**) clay, (**b**) sand, (**c**) silt.

### 3.5. Spatial Prediction of Soil Properties

The modeling results for each soil texture parameter were generalized to the entire study area, and prediction maps with a spatial resolution of 30 × 30 m were generated using ArcGIS 10.8 software. Figure 11 illustrates the prediction maps for all three models for each soil texture property.

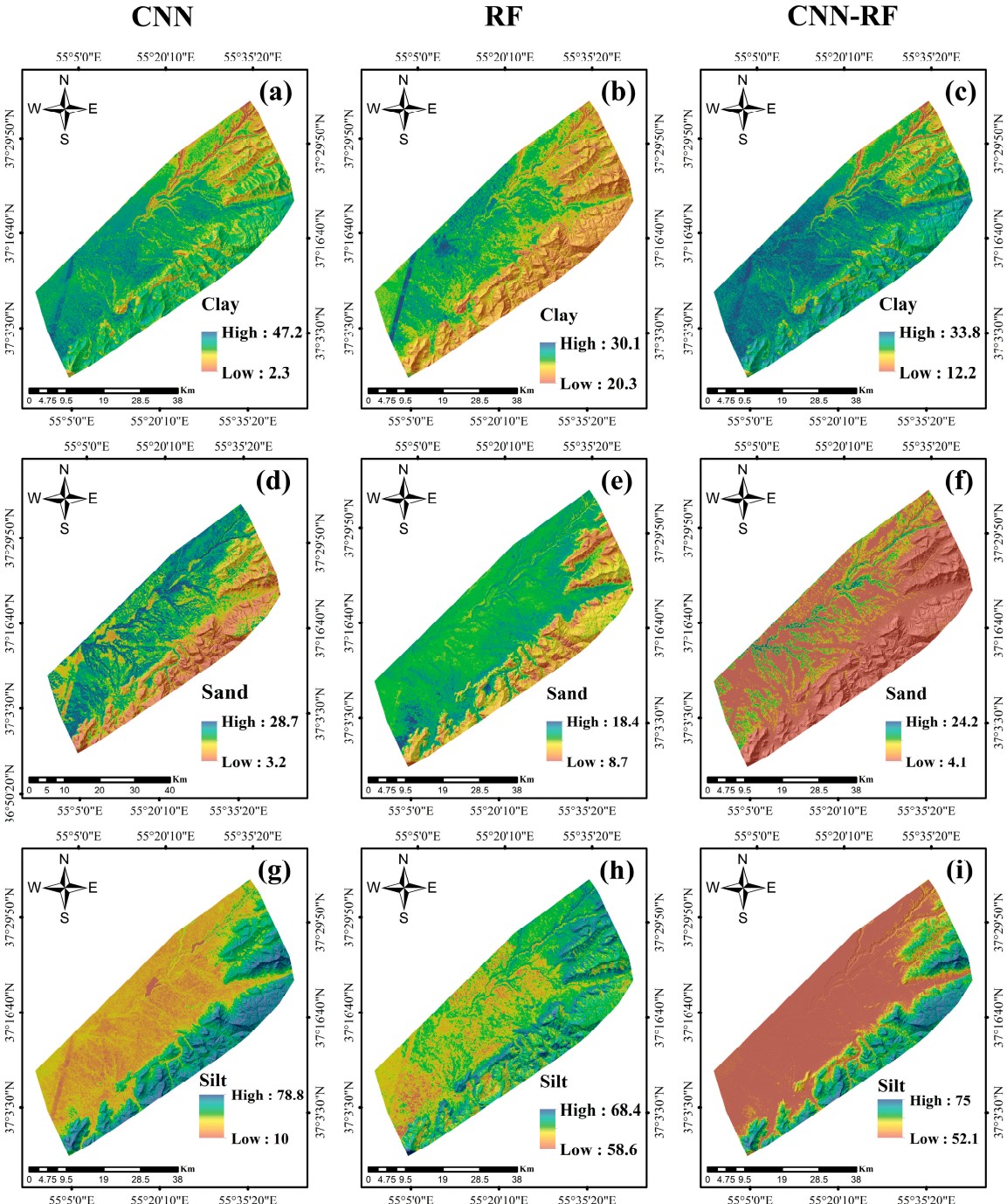

**Figure 11.** Digital maps of soil properties: (**a**–**c**) clay, (**d**–**f**) sand, (**g**–**i**) silt.

The amount of clay in the RF model prediction map decreases with increasing elevation. The prediction maps of the CNN and CNN-RF models are similar, with the central and southwestern points exhibiting higher amounts of clay in the CNN and CNN-RF models, respectively. The range of variation in clay content in the RF prediction map is smaller than that of the other models.

The prediction maps of the RF and CNN models are generally similar, except that the CNN map predicts slightly higher amounts of sand in the southwest and central regions than the RF model. Additionally, the range of sand content variation in the CNN-RF model prediction map is closer to the range of observed values compared to other models.

The amount of silt in the CNN prediction map decreases with decreasing elevation. In contrast to the RF model, the percentage of sand in the soil of the studied area does not exhibit a transparent relationship with elevation. In the prediction map of the hybrid CNN-RF model, the silt content is the highest in small parts of the southern and northeastern parts of the study area.

Overall, the range of texture fractions for all soil texture properties is closer to the maximum and minimum of actual values in the three prediction maps of the CNN-RF model compared to the stand-alone models. Additionally, there are more similarities between the prediction maps of RF and CNN than between the CNN-RF and RF models or between the CNN-RF and CNN models, except for clay, which exhibits slightly more similarity between the CNN-RF and CNN models.

Soil samples that were not utilized during the training phase were employed for the purpose of external validation. The soil texture maps were generated, and then the values extracted from each soil map were compared with the corresponding observed values to calculate the MSE. The evaluation results are presented in Table 7.

**Table 7.** Maps evaluation result.

| Properties | Models | MSE (%) |
|---|---|---|
| Clay | CNN | 0.076 |
| | RF | 0.0679 |
| | CNN-RF | 0.1027 |
| Sand | CNN | 0.095 |
| | RF | 0.094 |
| | CNN-RF | 0.078 |
| Silt | CNN | 0.178 |
| | RF | 0.137 |
| | CNN-RF | 0.569 |

The statistical assessment of predicted values for each soil property across various land cover categories is documented in Tables 8–10.

**Table 8.** The statistical parameters of the modeled soil texture on agricultural and forest land.

| Properties | Models | Agricultural Areas | | | | Forest Land | | | |
|---|---|---|---|---|---|---|---|---|---|
| | | Min | Max | Mean | Std | Min | Max | Mean | Std |
| Clay | CNN | 0.00 | 47.24 | 32.13 | 2.73 | 0.00 | 41.72 | 31.61 | 2.00 |
| | RF | 0.00 | 30.06 | 25.52 | 1.42 | 0.00 | 29.02 | 23.10 | 1.04 |
| | CNN-RF | 0.00 | 33.80 | 30.99 | 2.19 | 0.00 | 33.80 | 30.77 | 1.69 |
| Sand | CNN | 3.3 | 28.7 | 27.6 | 2.4 | 3.20 | 28.70 | 25.50 | 3.33 |
| | RF | 0.00 | 17.47 | 11.70 | 0.46 | 0.00 | 18.30 | 10.64 | 0.80 |
| | CNN-RF | 0.00 | 22.93 | 5.07 | 1.32 | 0.00 | 23.41 | 4.15 | 0.47 |
| Silt | CNN | 0.00 | 72.21 | 49.26 | 3.71 | 0.00 | 78.80 | 64.72 | 4.31 |
| | RF | 0.00 | 67.96 | 63.04 | 1.45 | 0.00 | 68.23 | 63.63 | 2.59 |
| | CNN-RF | 0.00 | 72.71 | 52.86 | 2.09 | 0.00 | 74.73 | 65.01 | 4.32 |

**Table 9.** The statistical parameters of the modeled soil texture on residential areas and uncovered plains.

| Properties | Models | Residential Areas | | | | Uncovered Plains | | | |
|---|---|---|---|---|---|---|---|---|---|
| | | Min | Max | Mean | Std | Min | Max | Mean | Std |
| Clay | CNN | 0.00 | 40.11 | 32.51 | 2.37 | 0.00 | 43.90 | 30.64 | 3.18 |
| | RF | 0.00 | 29.92 | 26.87 | 1.26 | 0.00 | 29.75 | 25.01 | 1.46 |
| | CNN-RF | 0.00 | 33.80 | 31.37 | 1.92 | 0.00 | 33.80 | 29.78 | 2.68 |
| Sand | CNN | 3.34 | 28.70 | 26.27 | 2.77 | 3.20 | 28.70 | 24.21 | 2.97 |
| | RF | 0.00 | 15.77 | 11.68 | 0.36 | 0.00 | 16.72 | 11.65 | 0.57 |
| | CNN-RF | 0.00 | 21.26 | 4.47 | 0.83 | 0.00 | 21.25 | 4.88 | 1.11 |
| Silt | CNN | 0.00 | 69.74 | 47.45 | 2.81 | 0.00 | 74.17 | 49.61 | 5.07 |
| | RF | 0.00 | 66.81 | 62.82 | 0.84 | 0.00 | 67.77 | 63.53 | 1.45 |
| | CNN-RF | 0.00 | 69.78 | 52.41 | 1.05 | 0.00 | 72.68 | 53.36 | 2.62 |

**Table 10.** The statistical parameters of the modeled soil texture on water bodies and range land.

| Properties | Models | Water Bodies | | | | Range Land | | | |
|---|---|---|---|---|---|---|---|---|---|
| | | Min | Max | Mean | Std | Min | Max | Mean | Std |
| Clay | CNN | 0.00 | 40.70 | 32.67 | 2.47 | 0.00 | 42.78 | 30.04 | 2.92 |
| | RF | 0.00 | 29.91 | 26.18 | 1.32 | 0.00 | 30.02 | 24.04 | 1.62 |
| | CNN-RF | 0.00 | 33.80 | 31.40 | 1.95 | 0.00 | 33.80 | 29.31 | 2.45 |
| Sand | CNN | 5.50 | 28.70 | 28.06 | 1.48 | 3.20 | 24.34 | 22.34 | 4.77 |
| | RF | 0.00 | 13.40 | 11.68 | 0.37 | 0.00 | 18.21 | 11.03 | 0.99 |
| | CNN-RF | 0.00 | 11.84 | 5.76 | 1.82 | 0.00 | 23.33 | 4.44 | 0.88 |
| Silt | CNN | 0.00 | 63.94 | 48.44 | 3.00 | 0.00 | 77.30 | 55.87 | 6.64 |
| | RF | 0.00 | 66.94 | 62.79 | 1.23 | 0.00 | 68.41 | 64.23 | 1.92 |
| | CNN-RF | 0.00 | 63.94 | 52.43 | 1.29 | 0.00 | 75.00 | 57.33 | 5.02 |

## 4. Discussion

### 4.1. Analysis of Parameters Affecting Soil Texture

In the RFE algorithm, the most influential parameter for soil clay was found to be MRVBF, which provides a better description of the region by identifying valley bottoms of various sizes and slopes [65]. MRVBF contains information about the location of the area that is directly related to clay [66], where the clay content increases from highlands to plains, similar to MRVBF.

For soil sand, the most effective parameter was found to be LST, which depends on the amount of solar energy absorbed by land cover types and local environmental conditions [67]. Sandy and agricultural areas absorb the highest temperatures due to the structure of the land cover [68]. The second parameter affecting sand was found to be B5, where areas with water dams had the highest amount of sand, and the amount of B5 reflection was the lowest. This is because water absorbs near-infrared the most [69], thus causing an opposite relationship between the B5 parameter and the amount of sand in the studied area.

The study's results indicate that B7 was the most important environmental parameter in predicting soil silt. SWIR bands play a crucial role in predicting and estimating soil texture properties, and particularly silt [70]. Furthermore, silt is one of the factors that influence the intensity of reflection and absorption of SWIR bands [71].

After determining feature importance, it was found that RS parameters had the most significant contribution to predicting each soil texture property. RS provides these parameters with proper spatial and temporal accuracy [72]. The selected RS parameters included seven parameters, including RI, B7, SI, CI, B5, NDVI, and CLI, with B7 and NDVI having the most significant influence. NDVI is one of the most widely used vegetation indices that reduces the influence of atmosphere and soil background in spectral measurements [5]. SI, CI, and RI are parameters extracted from the three visible bands of Landsat 8 (B2, B3, and B4) (Table 4) and have a significant impact on predicting soil properties [73]. These parameters are obtained from Landsat 8 data with advantages such as short periodicity, good spatial resolution and coverage, and a wide range of spectral ranges including visible, B5, and SWIR [74]. Several studies have demonstrated that incorporating RS variables improves prediction accuracy [40,75,76].

*4.2. Model Comparison and Analysis*

Based on the findings, the hybrid CNN-RF model exhibited greater precision compared to the individual CNN and RF models. The convolutional layers employed at the outset of the modeling process enabled the extraction and organization of input data features [77]. Conversely, a CNN's employment of a fully connected layer for the final regression decision often leads to overfitting [78]. Consequently, incorporating the RF algorithm enhanced the accuracy of the results [79]. In recent years, numerous studies have applied CNN and RF algorithms across different domains. For instance, the fused CNN-RF model has been employed to detect electricity theft [80], yielding improved accuracy in comparison to the individual CNN and RF models. Furthermore, for tree species classification, a fusion of CNN and RF algorithms outperformed stand-alone CNN, SVM, and RF models [81]. Additionally, for product classification using satellite images, the combined one-dimensional CNN approach with RF achieved greater accuracy in contrast to the CNN-1D networks and the fused LSTM-RF network [78]. Li et al. (2022) demonstrated the superiority of the CNN-RF hybrid model for estimating actual evapotranspiration compared to the CNN-SVM and individual CNN an RF models [82].

*4.3. Strengths and Weaknesses*

The current study exhibits several strengths, including the hybridization of the RF ML algorithm and CNN DL neural network for the spatial prediction of soil texture, leading to increased accuracy compared to the individual CNN and RF models. Moreover, the use of RS data has enabled the extraction of multiple variables that influence soil texture at a suitable scale and with reduced costs. However, the lack of soil samples at high altitudes and the use of feature importance instead of a meta-heuristic algorithm or the wrapper method for feature selection are limitations of this research.

## 5. Conclusions and Recommendations

The objective of the present study was to compare and evaluate the performance of CNN, RF, and CNN-RF algorithms for spatial prediction of soil texture properties. Satellite images were employed due to their appropriate spatial and temporal accuracy in preparing indicators that impact soil texture. The study yielded the following outcomes: (1) The RF algorithm identified MRVBF, LST, and B7 as the most effective parameters for clay, sand, and silt, respectively. (2) Among the effective parameters, the RS variables had the largest contribution to the modeling input. Specifically, NDVI, B7, SI, B5, CI, RI, and CLI were found to be the critical RS parameters influencing soil texture. (3) The hybrid CNN-RF model demonstrated the highest accuracy in predicting soil texture properties, as indicated by the evaluation results. (4) Sand, silt, and clay exhibited greater accuracy based on the MSE evaluation metric.

The prediction maps generated via the hybrid CNN-RF model can aid agricultural management, soil erosion monitoring, and irrigation. Potential areas for future research include: (1) Utilizing a meta-heuristic algorithm in lieu of the RFE algorithm to improve

modeling accuracy. (2) Extracting variables such as homogeneity, contrast, dissimilarity, and entropy in the studied area using the gray-level cooccurrence matrix to enhance soil texture prediction accuracy. (3) Exploring the integration of additional ML and DL models.

**Author Contributions:** Conceptualization, F.S.H. and S.V.R.-T.; Data curation, F.S.H. and M.J.; Formal analysis, F.S.H., S.V.R.-T. and M.B.S.; Funding acquisition, A.S.-N. and S.-M.C.; Investigation, S.V.R.-T.; Methodology, F.S.H. and M.B.S.; Project administration, A.S.-N. and S.-M.C.; Resources, M.B.S.; Software, F.S.H. and M.B.S.; Supervision, A.S.-N. and S.-M.C.; Validation, M.B.S. and M.J.; Visualization, F.S.H.; Writing—original draft, F.S.H.; Writing—review and editing, S.V.R.-T., M.B.S., A.S.-N., M.J. and S.-M.C. All authors have read and agreed to the published version of the manuscript.

**Funding:** This work was supported in part by an Institute of Information and communications Technology Planning and Evaluation (IITP) grant funded by the Korea government (MSIT) (no. IITP-2023-RS-2022-00156354), in part by the Ministry of Trade, Industry, and Energy (MOTIE) and the Korea Institute for Advancement of Technology (KIAT) (no. P0016038), and in part by a National Research Council of Science and Technology (NST) grant by the Korea government (MSIT) (No. CRC21011).

**Institutional Review Board Statement:** Not applicable.

**Informed Consent Statement:** Not applicable.

**Data Availability Statement:** The data that support the findings of this study are available from the corresponding author, Soo-Mi Choi, upon reasonable request.

**Conflicts of Interest:** The authors declare no conflict of interest.

## Appendix A

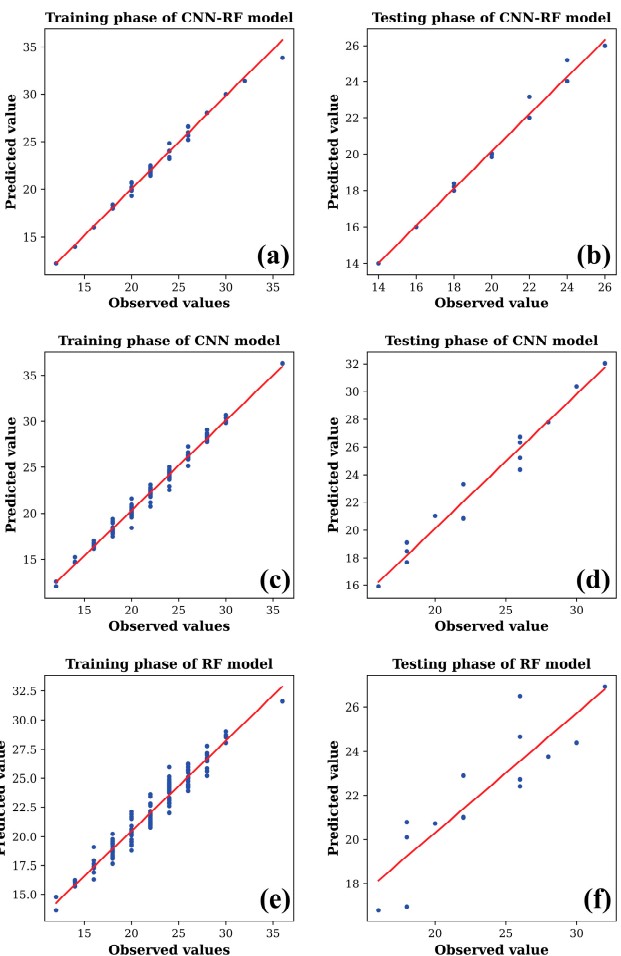

**Figure A1.** Scatter plots of all the proposed models for clay: (**a**,**b**) CNN-RF, (**c**,**d**) CNN, (**e**,**f**) RF.

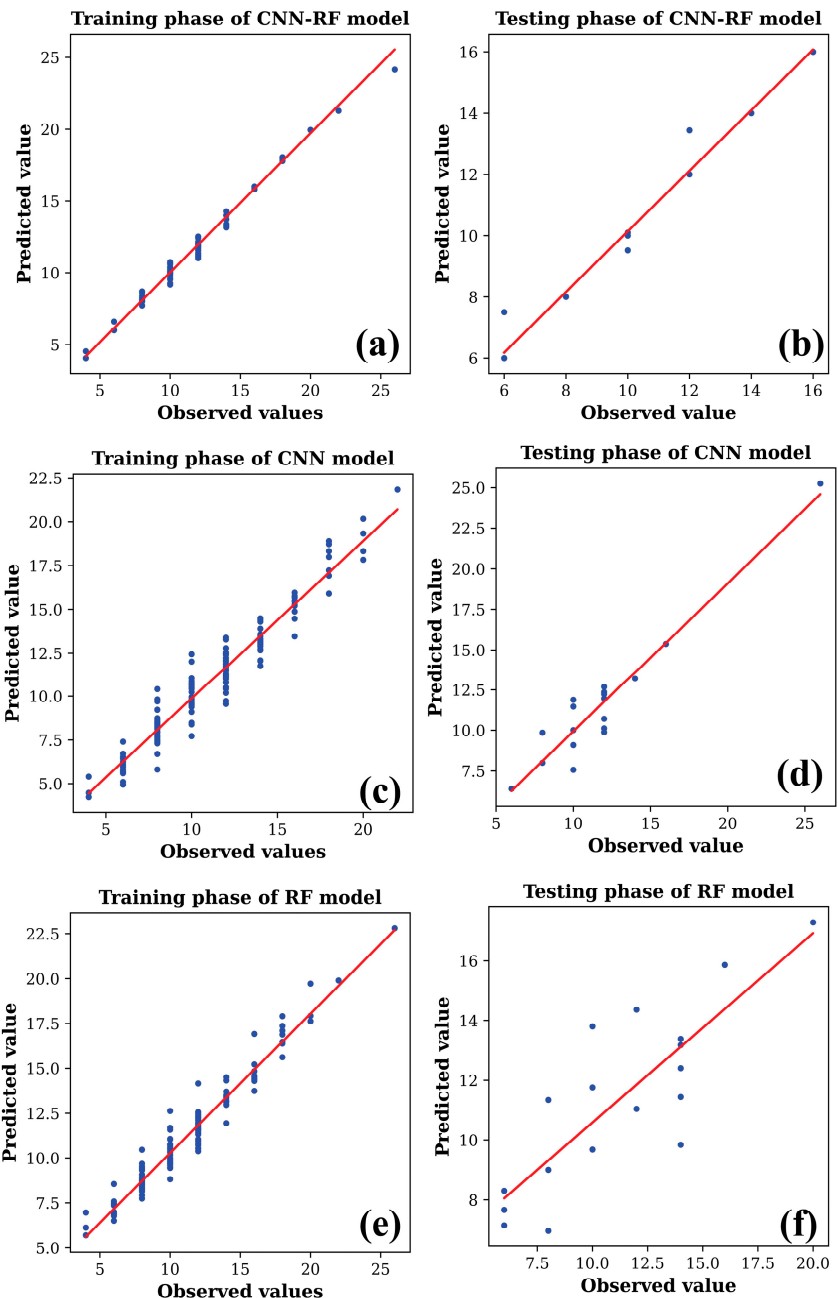

**Figure A2.** Scatter plots of all the proposed models for sand: (**a**,**b**) CNN-RF, (**c**,**d**) CNN, (**e**,**f**) RF.

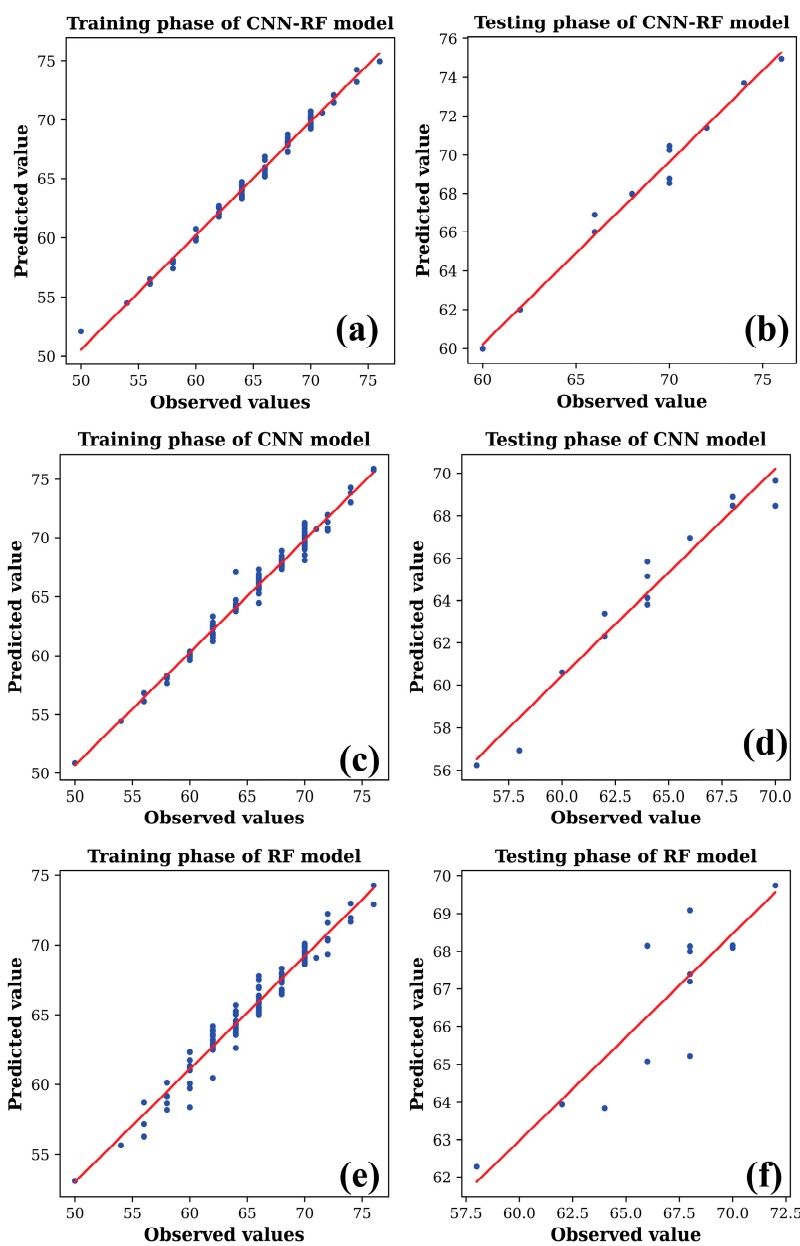

**Figure A3.** Scatter plots of all the proposed models for silt: (**a**,**b**) CNN-RF, (**c**,**d**) CNN, (**e**,**f**) RF.

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
