# Peer review of "Geospatial Artificial Intelligence (GeoAI) and Satellite Imagery Fusion for Soil Physical Property Predicting"

_sustainability, doi:10.3390/su151914125_

Round 1

Reviewer 1 Report

The research addresses a significant issue in agriculture and environmental protection by predicting soil physical properties.

The following points need to be addressed.

In this article the authors predicted the properties of the soil. may I know why they selected regression problem instead of classification?

2.5. Models Evaluation - Needs to be verified. Because based on the attribute / features, please state about the target / dependent variable?

In this article, the authors should clarify whether clay or sand or silt will be the target or whether it contains any range?

in my point of view, it looks like classification problem not a regression problem? Can you clarify this?

In such a case does the evaluation metric and results need to be changed? Because if it is a regression problem, what authors are predicting? 

Minor editing of English language required

Author Response

#Please see the attachment.#

We are grateful to the reviewers for their kind help and constructive comments which helped improving the presentation of the paper. Thank you for the reviewers ‘comments concerning our manuscript entitled “Geospatial Artificial Intelligence (GeoAI) and Satellite Imagery Fusion for Soil Physical Property predicting” (ID: sustainability-2543879). Those comments are all valuable and very helpful for revising and improving our paper, as well as the important guiding significance to our researches. We have studied comments carefully and have made correction which we hope meet with approval. We hope that the revised manuscript is to your satisfaction. We are more than happy to improve the paper again according to new comments and suggestions that might come. Please note that in some places of the revised manuscript, we have made improvements in addition to your comments. All the questions of the respected reviewer were answered point by point. The next section contains our point-by-point responses (in blue) and changes referencing the manuscript (in green), based on the reviewers’ comments (in italic).

Comments and Suggestions for Authors:

The research addresses a significant issue in agriculture and environmental protection by predicting soil physical properties.

The following points need to be addressed.

  1. In this article, the authors predicted the properties of the soil. may I know why they selected regression problem instead of classification?

Response:

We appreciate your interest in understanding our choice of a regression problem for predicting soil properties in this study. The decision to opt for regression was driven by several considerations. In classification scenarios, the target variables represent "YES" and "NO," often denoted as one and zero. In contrast, regression problems involve continuous variables, such as soil properties like silt content ranging from a minimum of 50 to a maximum of 76 percentage. Soil texture properties (sand, silt, and clay) are continuous variables. Regression is well-suited for modeling such continuous outcomes, allowing us to predict precise numerical values for these properties. In addition, our study aimed to establish quantitative relationships between various input factors, such as remote sensing data, topographic parameters, climatic parameters, and the corresponding soil properties. Regression analysis provides a robust framework to quantify these relationships and understand how changes in the input variables affect the continuous outcomes of soil properties.

In summary, our choice to employ regression rather than classification was guided by the need to accurately predict continuous soil property values, quantify relationships, and provide detailed insights into the influencing factors.

  1. Models Evaluation - Needs to be verified. Because based on the attribute / features, please state about the target / dependent variable?

Response:

We appreciate your interest in clarifying the evaluation of our models. The target or dependent variable in our study refers to soil properties, specifically the content percentages of sand, silt, and clay.  These properties are the variables our models are designed to predict. The evaluation process involves assessing the accuracy of the predicted values for these soil properties based on the input attributes or features, which include remote sensing data, topographic parameters, and climatic variables. In our study, we employed Mean Squared Error (MSE), Root Mean Squared Error (RMSE), and the coefficient of determination (R2) as evaluation metrics to assess the performance of our models [1-3].

2.5. Models Evaluation

The efficiency of the model is evaluated using three metrics: Root Mean Squared Error (RMSE), Mean Squared Error (MSE), and coefficient of determination (R2) (Equations 4-6). Lower MSE and RMSE values indicate higher modeling accuracy. R2 illustrates the goodness of fit between the data and the regression model. The value of R2 ranges from 0 to 1, with values closer to 1 indicating better model performance [4, 5].

(4)

(5)

(6)

In these equations,   represents the measured value,  represents the predicted value,  represents the mean of the actual values, and  is the number of observations. Another effective way to display the relationship between statistical indicators and to visualize the difference in model performance in predicting soil properties is to use a Taylor diagram [6]. Taylor diagrams show the degree of agreement between predicted and observed values regarding correlation and standard deviation error. Additionally, a box plot is used to compare the minimum and maximum values of the range, the upper and lower quartiles, and the median of the predicted values and the actual data. This set of values provides a concise summary of the distribution of the dataset [7].

  1. In this article, the authors should clarify whether clay or sand or silt will be the target or whether it contains any range?

Response:

Thank you for your comment. In machine and deep learning algorithms, there exists a singular target variable. The primary goal of these models is to predict this specific target variable. Our study is centered around predicting soil properties using these methods. In our study, we have categorized the parameters into two distinct groups: dependent parameters, which represent soil texture properties, and independent parameters, encompassing various environmental parameters. The dependent parameters are treated as target variables, and the models are designed to predict each of the three soil properties individually. There are two reasons why we chose to predict these soil properties separately. Firstly, each soil property has a different range of target values. Secondly, by predicting each soil property separately, we can obtain more accurate and specific predictions for each property.

The table provided below displays an example of the dataset that was utilized as input for models:

Independent parameters

Target

B1

B2

CI

LST

Aspect

Elevation

DR

Clay

11402

10837

11200

11725

17365

18314

14704

22

11752

11193

10984

11244

15226

16016

13407

16

13036

12946

13994

15437

19529

21470

18202

14

10982

10157

10066

9327

14307

13143

9669

14

12189

12024

12318

13627

17899

19170

15123

22

Environmental parameters that affect each soil property were mentioned in the manuscript. The context of “Environmental Parameters” section updated as follows to address your concern.

Based on previous studies [8-10], expert opinions, and the specific conditions of the studied area, three groups of environmental parameters were used. These included remote sensing variables such as Band 1 (B1) to Band 5 (B5) and Band 7 (B7) of Landsat-8, Brightness index (BI), Coloration index (CI), Clay index (CLI), Enhanced vegetation index (EVI), Land surface temperature (LST), Hue index (HI), Normalized difference vegetation index (NDVI), Redness index (RI), and Saturation index (SI). Climate variables such as air temperature and rainfall were also included, along with topographic variables including aspect, elevation, slope, duration radiation (DR), multi-resolution index of valley bottom flatness (MRVBF), multi-resolution ridgetop flatness index (MRRTF), and topographic wetness index (TWI). In this study, dependent parameters represent soil texture properties, which are treated as target variables, and independent parameters encompass various environmental parameters.

Table 3. Illustrates the parameters that impact the properties of soil texture.

Number of parameters

Effective parameters

Soil Texture

18

NDVI, Elevation, B7, B5, B1, B2, B3, B4, MRRTF, MRVBF,

 Rainfall, SI, CI, LST, Temp, Aspect, RI, TWI

Clay

18

NDVI, Elevation, B7, B5, B3, B4, MRRTF, MRVBF, 

SI, BI, CLI, CI, Slope,EVI, DR, Aspect, RI, TWI

Silt

18

NDVI, Elevation, B7, B5, B1, B2, B3, B4, Rainfall, SI, BI, CLI, 

MRRTF, MRVBF, CI, Slope, LST, DR

Sand

  1. In my point of view, it looks like classification problem not a regression problem? Can you clarify this? In such a case does the evaluation metric and results need to be changed? Because if it is a regression problem, what authors are predicting

Response:

Thank you for your comment. While it may seem like a classification problem at first glance, our study actually is a regression problem. Our primary objective was to predict the quantitative values of specific soil texture properties, namely clay, sand, and silt content, based on various remote sensing, topographic, and climatic parameters. The minimum and maximum of soil properties are presented in Table 2 in manuscript. These properties are continuous variables.

Table 2. Statistical summary of soil texture after outlier removal

Sand (%)

Silt (%)

Clay (%)

Soil Texture

4

50

12

Minimum

26

76

36

Maximum

11.056

65.74

22.182

Mean

3.705

4.567

4.199

Standard deviation

In a classification problem, the goal is to assign instances to predefined classes or categories, whereas in our case, we aimed to predict continuous numerical values that represent the proportions of different soil texture properties. We treated the problem as a regression task since the output we are predicting is a continuous range of values, and our analysis focused on developing models that can provide accurate quantitative estimates of these properties. The final maps are shown in figure 11. The legends within Figure 11 highlight that the outputs are continuous variables. In the case of a classification problem, the output would have included two classes.

Figure 11. Digital maps of soil properties: a, b, c) Clay, d, e, f) Sand, g, h, i) Silt

In response to the second part of your comment, we would like to explain that while classification problems often use metrics like accuracy, precision, and recall, regression problems typically employ metrics such as Mean Squared Error (MSE), Root Mean Squared Error (RMSE), and R-squared (R2) to assess the predictive performance. To address your concern, there is no need to change the evaluation metrics. Evaluation results corresponding to each soil property and model are presented in Table 6.

Table 6. Evaluation results

Runtime (s)

Test

Train

Models

Properties

RMSE (%)

MSE (%2)

RMSE (%)

MSE (%2)

2.67

0.966

0.019

0.00038

0.981

0.013

0.00016

CNN

Clay

0.23

0.636

0.064

0.00407

0.910

0.028

0.00079

RF

0.21

0.982

0.010

0.00010

0.995

0.007

0.00005

CNN-RF

1.36

0.908

0.022

0.00046

0.928

0.017

0.00029

CNN

Sand

0.44

0.683

0.037

0.00135

0.917

0.018

0.00034

RF

0.29

0.976

0.008

0.00007

0.992

0.006

0.00003

CNN- RF

2.73

0.913

0.020

0.00040

0.920

0.016

0.00024

CNN

Silt

0.196

0.676

0.024

0.00060

0.935

0.015

0.00022

RF

0.215

0.980

0.010

0.00009

0.987

0.006

0.00004

CNN- RF

[1]           S. Mantena, V. Mahammood, and K. N. Rao, "Prediction of soil salinity in the Upputeru river estuary catchment, India, using machine learning techniques," Environmental Monitoring and Assessment, vol. 195, no. 8, p. 1006, 2023.

[2]           S. Saidi, S. Ayoubi, M. Shirvani, K. Azizi, and M. Zeraatpisheh, "Comparison of Different Machine Learning Methods for Predicting Cation Exchange Capacity Using Environmental and Remote Sensing Data," Sensors, vol. 22, no. 18, p. 6890, 2022.

[3]           A. Usta, "Prediction of soil water contents and erodibility indices based on artificial neural networks: using topography and remote sensing," Environmental Monitoring and Assessment, vol. 194, no. 11, p. 794, 2022.

[4]           S. V. Razavi-Termeh, A. Sadeghi-Niaraki, and S.-M. Choi, "Spatial modeling of asthma-prone areas using remote sensing and ensemble machine learning algorithms," Remote Sensing, vol. 13, no. 16, p. 3222, 2021.

[5]           M. Farahani, S. V. Razavi-Termeh, and A. Sadeghi-Niaraki, "A spatially based machine learning algorithm for potential mapping of the hearing senses in an urban environment," Sustainable Cities and Society, vol. 80, p. 103675, 2022.

[6]           K. E. Taylor, "Summarizing multiple aspects of model performance in a single diagram," Journal of geophysical research: atmospheres, vol. 106, no. D7, pp. 7183-7192, 2001.

[7]           K. Potter, H. Hagen, A. Kerren, and P. Dannenmann, "Methods for presenting statistical information: The box plot," in VLUDS, 2006, pp. 97-106.

[8]           S. Fathololoumi, A. R. Vaezi, S. K. Alavipanah, A. Ghorbani, D. Saurette, and A. Biswas, "Improved digital soil mapping with multitemporal remotely sensed satellite data fusion: A case study in Iran," Science of the Total Environment, vol. 721, p. 137703, 2020.

[9]           M. Shahriari, M. Delbari, P. Afrasiab, and M. R. Pahlavan-Rad, "Predicting regional spatial distribution of soil texture in floodplains using remote sensing data: A case of southeastern Iran," Catena, vol. 182, p. 104149, 2019.

[10]         R. Taghizadeh-Mehrjardi, H. Khademi, F. Khayamim, M. Zeraatpisheh, B. Heung, and T. Scholten, "A comparison of model averaging techniques to predict the spatial distribution of soil properties," Remote Sensing, vol. 14, no. 3, p. 472, 2022.

Reviewer 2 Report

This article reports the application of neuronal network models towards the
improvement of Digital Soil Mapping (DSM) methods. In particular, the authors
set out to study the effectiveness of convolutional neuronal networks (CNN)
as means of feature elimination. This is a welcomed study in concept, however
its practical realisation does not provide for a successful evaluation. The
dataset selected for the exercise is composed of mere three hundred observed
values and four dozen features. In contrast, CNN are recognised as successful
with dozens of thousands of observed values and features numbered in the
hundreds. In mundane terms, this study tried to prove a fly can by killed with
an elephant gun. The performance indicators clearly show this to be the case,
with marginal (and spuriously different) errors.

Besides the overall inadequacy of the realisation to the concept, there are
serious methodology flaws to consider:

- No reasoning is provided for the selection of such a sophisticated model as
  CNN.

- No attempt is made to compare the CNN feature selection method with
  traditional feature selection methods.

- Texture fractions are taken as independent variables and predicted
  independently, resulting in unusable soil maps.

- The cross-validation procedure is not properly described, and its application
  referred ambiguously. Validation folds are first described as being fully used
  in training, but latter out-of-bag testing is referred.

- No investigation is made on the reasons for the diminutive error metrics
  obtained. Spurious error differences are classified as "great".

Detailed comments to the manuscript can be found in the PDF file attached.

Considering the above, I am obliged to recommend the rejection of this study in
its current form. However, I would still encourage the authors to further
pursue this research, with the following recommendations:

- Test the concept on large dataset, with dozens of thousands of soil
  observation results and a much larger and heterogeneous spatial area.

- Start by targeting independent soil properties such as pH or Nitrogen
  content.

- Test simpler NN architectures ahead of the sophisticated CNN.

Author Response

#Please see the attachment.#

We are grateful to the reviewers for their kind help and constructive comments which helped improving the presentation of the paper. Thank you for the reviewers ‘comments concerning our manuscript entitled “Geospatial Artificial Intelligence (GeoAI) and Satellite Imagery Fusion for Soil Physical Property predicting” (ID: sustainability-2543879). Those comments are all valuable and very helpful for revising and improving our paper, as well as the important guiding significance to our researches. We have studied comments carefully and have made correction which we hope meet with approval. We hope that the revised manuscript is to your satisfaction. We are more than happy to improve the paper again according to new comments and suggestions that might come. Please note that in some places of the revised manuscript, we have made improvements in addition to your comments. All the questions of the respected reviewer were answered point by point. The next section contains our point-by-point responses (in blue) and changes referencing the manuscript (in green), based on the reviewers’ comments (in italic).

Comments:

  1. How is this approach innovative? None of the methods described here are new to digital soil mapping. (line 19)

Response:

We appreciate the reviewer's engagement with our work and their question regarding the innovation of our approach in the context of digital soil mapping. While it's true that the individual methods we employed in our study may not be entirely new to the field of digital soil mapping, our contribution lies in the innovative fusion and integration of these methods to improve soil physical property predictions. Firstly, we innovatively hybridized the RF and CNN models, a novel approach in soil-related studies. Secondly, we extended our analysis by incorporating a comprehensive range of remote sensing (RS) indices, which sets our study apart. Although both RF and CNN were implemented in digital soil mapping, our study introduces a unique application of this hybrid approach to predict soil texture properties, which has not been extensively explored in the context of digital soil mapping. By combining the strengths of RF and CNN models, we aimed to capitalize on their complementary capabilities and achieve higher prediction accuracy. This integration of methodologies brings a fresh perspective to the field and contributes to advancing soil texture prediction techniques.

  1. What are these? (line 36)

Response:

Thank you for your opinion. MRVBF, LST, and B7 represent the acronyms for multi-resolution index of valley bottom flatness (MRVBF), Land surface temperature (LST), and Band 7 (B7) of Landsat-8 satellite imagery, respectively. These environmental parameters were identified by the RF algorithms as the most significant factors influencing clay, sand, and silt, respectively.

  1. Perhaps, but with only 317 samples your dataset is really small. (line 74)

Does your dataset classify as such? (line 79)

Response:

Thank you for bringing up this point. In the introduction section, our intention was to emphasize the broader benefits of employing deep learning models, rather than directly addressing the scale or complexity of our dataset. However, it's worth noting that the dataset consists of 317 rows representing soil samples and 18 columns containing various features. This dataset configuration, which involves 317 samples with 18 distinct attributes, does exhibit a certain level of intricacy. Moreover, the study area spans 2237 columns and 2435 rows, with each pixel measuring 30 * 30 meters. Our aim is to predict soil properties for the entire study area, and thus the input matrix for generating predictive maps encompasses the entirety of this spatial extent. In fact, in the first stage, the model is fitted with 317 samples, but the final goal of the research is to fit this model on a large dataset, which is the study area. In fact, our final dataset is a combination of 2237 columns and 2435 rows for 18 different criteria, which is relatively an acceptable dataset for deep learning.

Furthermore, our emphasis on fusion methods and the integration of multiple data sources, including satellite imagery and ground-based measurements, adds another layer of complexity to the modeling process. The careful integration of these diverse data sources demands a nuanced approach to model training and validation.

  1. In general, this is not true. The performance of these models is rather tied to the heterogeneity of data. (lines 85-86)

Response:

We appreciate the reviewer's input and would like to address their point regarding the performance of machine learning (ML) algorithms in relation to data volume and heterogeneity. We acknowledge that the statement made in the paper might have been overly generalized and could benefit from further clarification. Indeed, the performance of ML algorithms can be significantly influenced by both the heterogeneity and the volume of data. While it's true that some ML algorithms can demonstrate saturation in performance as data volume increases, this observation is not universal and depends on several factors, including the complexity of the model, the nature of the data, and the algorithm's inherent capacity to handle larger datasets.

In cases where the data is highly heterogeneous and contains valuable information across various scales and contexts, ML algorithms often stand to benefit from an increased volume of diverse data. This diversity can enable the models to better capture complex patterns and relationships within the data, leading to improved predictive performance.

Conversely, there are scenarios where the addition of more data might not necessarily enhance model performance if the new data lacks relevance or doesn't introduce meaningful variation to the existing dataset. It's essential to carefully curate and preprocess the data to ensure that any increase in data volume contributes positively to model training and generalization.

In our specific study, we've taken into account the balance between data volume and heterogeneity. The fusion of multiple data sources, including satellite imagery and ground-based measurements, aims to provide a rich and diverse dataset that can effectively capture the complex spatial and spectral patterns of soil properties. We also recognize that while the performance of ML algorithms might initially improve with additional data, there can be diminishing returns if the data lacks informative diversity.

In light of these considerations, we will revise the statement in the paper to better reflect the nuanced relationship between ML algorithm performance, data volume, and data heterogeneity. We appreciate the reviewer's insight, which helps to refine the accuracy of our presentation.

Revised as follow:

While some ML algorithms may exhibit saturation in performance as data volume increases, the relationship between data volume and algorithm performance is influenced by various factors, including the heterogeneity and relevance of the data. In cases where data is diverse and contains valuable information across different scales and contexts, in-creasing data volume can enhance model performance. However, it's essential to carefully curate and preprocess the data to ensure that the additional volume contributes meaningfully to model training and generalization.

  1. This assertion fails if data heterogeneity increases with volume. (lines 86-87)

Response:

Thank you for your informative feedback. In line with your input, we have included the following sentence which aligns more closely with our data and methodology.

While some ML algorithms may exhibit saturation in performance as data volume increases, the relationship between data volume and algorithm performance is influenced by various factors, including the heterogeneity and relevance of the data. In cases where data is diverse and contains valuable information across different scales and contexts, in-creasing data volume can enhance model performance. However, it's essential to carefully curate and preprocess the data to ensure that the additional volume contributes meaningfully to model training and generalization. Moreover, ML algorithms cannot detect irrelevant and redundant information, negatively impacting their performance [17]. While ML can handle complex data, excessive hidden layers can lead to issues like overfitting and vanishing gradients [19, 20]. DL, with its strong predictive accuracy, outperforms ML in spatial prediction. To tackle intricate soil challenges, sophisticated algorithms like Convolutional Neural Network (CNN), rooted in DL, are used to boost accuracy and re-duce uncertainty [21, 22].

  1. This research is unique in two significant ways: firstly, it proposes a hybrid CNN-RF model for texture prediction, and secondly, it employs optical satellite images to enhance the accuracy of predictions. How is this aspect innovative?(line 116)

Response:

 Thank you for your feedback. This research introduces uniqueness in two main aspects: firstly, it presents a novel hybrid CNN-RF model for soil texture prediction; Secondly, it leverages the power of optical satellite images, particularly Landsat 8 data, along with additional datasets like SRTM, to enhance the precision of the predictive models. This integration of diverse remote sensing indices and image data significantly contributes to the accuracy of the predictions. While it's true that the individual methods we employed in our study may not be entirely new to the field of digital soil mapping, our contribution lies in the innovative fusion and integration of these methods to improve soil physical property predictions.

  1. This is very smaller area for this kind of exercise, i.e. digital soil mapping with remote sensing products. Would be important to declare from the onset the target spatial resolution. (lines 123-124)

Response:

We appreciate the reviewer's observation and the importance they place on explicitly declaring the target spatial resolution in the context of digital soil mapping with remote sensing products. We agree that defining the intended spatial resolution from the outset is crucial for understanding the scope and implications of our study. In our pursuit of accurately predicting soil physical properties using the fusion of various methods and data sources, we indeed recognize that the study area's spatial extent might be perceived as relatively small for exercises involving digital soil mapping and remote sensing products. The area's size directly influences the achievable spatial resolution and the level of detail that can be captured in the predictive maps. To address this concern and provide greater clarity, we will revise the paper to explicitly state the target spatial resolution in the introduction section. By doing so, we aim to establish clear expectations regarding the scale of our study and the resolution at which our predictions are generated. This addition will help readers better understand the limitations and opportunities presented by the scale of our exercise in the context of digital soil mapping and remote sensing products.

Based on the opinion of the respected reviewer, it was added to the part of the study area as follows:

Our study aims to predict soil physical properties using a fusion of methods and data sources, focusing on the unique challenges the study area poses. The target spatial resolution for our predictions is 30*30 m, which reflects the scale at which we aim to generate predictive maps.

  1. The highest and 130 lowest elevations in the area are 1722 meters above sea level, respectively. And the lowest? (line 131)

Response:

Thank you for your comment. The updated manuscript now states:

"The highest and lowest elevations in the area are 1722 and zero meters above sea level, respectively."

  1. The distribution of samples appears way too close to justify the application of remote sensing products. Again, information on the target spatial resolution would be necessary.

Response:

We appreciate the reviewer's comment and their concern regarding the distribution of samples in relation to the application of remote sensing products. We acknowledge the importance of ensuring an appropriate sample distribution to effectively justify utilizing remote sensing data. To address this concern and provide the necessary information, we will enhance the paper by providing explicit details about the target spatial resolution in the relevant sections (Study area and soil sample sections). In fact, in the study area, it was not possible to collect samples in mountainous areas, so most of the samples were collected in agricultural areas.

  1. More than one grid was applied? There might an error in this sentence.(line 140)

Response:

Thank you for your comment. The study utilized a grid sampling approach to collect soil samples. This method involves dividing the study area into a grid pattern, and each point where the grid intersects represents a potential sampling location. In our case, the sampling area was divided into multiple grids, each covering an area of 1 km2. This was done to ensure comprehensive coverage of the study area and capture potential spatial variations in soil properties. The term "grid" in this context refers to the division of the study area into smaller sections for systematic sampling.

  1. Is this a sample per site/location? At which depth were the samples collected? And where is the dataset available? Line(141)

Response:

Thank you for your feedback. The samples were collected at a depth of 0 to 30 cm. Each sampling location involved taking five samples: one at the main point and four at a 10 m distance from the main point. These samples were mixed to create a representative composite for that specific point.

  1. What are the units of these values.(line 153, line 158)

Response:

Thank you for your comment. The measured data in our study is presented in percentage (%).

Table 1. Statistical summary of soil texture

Sand (%)

Silt (%)

Clay (%)

Soil Texture

0

0

0

Minimum

58

80

44

Maximum

12.867

64.457

22.322

Mean

9.115

9.249

6.920

Standard deviation

Table 2. Statistical summary of soil texture after outlier removal

Sand (%)

Silt (%)

Clay (%)

Soil Texture

4

50

12

Minimum

26

76

36

Maximum

11.056

65.74

22.182

Mean

3.705

4.567

4.199

Standard deviation

  1. Samples or points? These terms have different meanings in soil science (e.g. ISO-28258). (line 156)

Response:

Thank you for addressing this matter. Considering that we collected five samples from 0 to 30 cm depth – one at the primary point and four at a 10 m distance from it – and then mixed them to create a representative sample for that point, it appears that our sampling method is consistent with the ISO 28258 standard's recommendation on composite sampling. In the context of our study, "samples" denote the collected soil specimens, while "points" signify the distinct geographical locations where these samples were obtained. In the revised manuscript, we used "soil samples" instead of "sample points"

  1. You started with 317 samples, removed outliers and ended up with 478 samples. Something is wrong here. (line 156)

Response:

Thank you for raising this concern. Initially, we began with a dataset containing 317 samples for each soil property. Subsequently, outlier removal was carried out using the Theil-Sen regression method. The numbers 179, 144, and 155 correspond to subsets derived from the initial 317 samples. These subsets were obtained by eliminating outliers from the dataset for each specific soil property, namely sand, silt, and clay.

  1. This table is not referenced in the text. Why are there different features listed for each variable? Sand, silt and clay are dependable variables, they cannot be predicted separately. (line 171)

Response:

Thank you for pointing this out. In the revised manuscript, the table is now properly referenced in the text. We appreciate your inquiry regarding the separate prediction of sand, silt, and clay. There are two reasons why we chose to predict these soil properties separately. Firstly, each soil property has a different range of target values, and predicting them simultaneously may increase the errors in the predictions. Secondly, by predicting each soil property separately, we can obtain more accurate and specific predictions for each property.

Revised as follow:

2.3. Environmental Parameters

Based on previous studies [5-7], expert opinions, and the specific conditions of the studied area, three groups of environmental parameters were used. These included RS variables such as Band 1 (B1) to Band 5 (B5) and Band 7 (B7) of Landsat-8, Brightness index (BI), Coloration index (CI), Clay index (CLI), Enhanced vegetation index (EVI), Land surface temperature (LST), Hue index (HI), Normalized difference vegetation index (NDVI), Redness index (RI), and Saturation index (SI). Climate variables such as air temperature and rainfall were also included, along with topographic variables including aspect, elevation, slope, duration radiation (DR), multi-resolution index of valley bottom flatness (MRVBF), multi-resolution ridgetop flatness index (MRRTF), and topographic wetness index (TWI). The environmental parameters that impact each soil property are listed in Table 3.

Table 3. Illustrates the parameters that impact the properties of soil texture.

Number of parameters

Effective parameters

Soil Texture

18

NDVI, Elevation, B7, B5, B1, B2, B3, B4, MRRTF, MRVBF,

 Rainfall, SI, CI, LST, Temp, Aspect, RI, TWI

Clay

18

NDVI, Elevation, B7, B5, B3, B4, MRRTF, MRVBF, 

SI, BI, CLI, CI, Slope, EVI, DR, Aspect, RI, TWI

Silt

18

NDVI, Elevation, B7, B5, B1, B2, B3, B4, Rainfall, SI, BI, CLI, 

MRRTF, MRVBF, CI, Slope, LST, DR

Sand

  1. The use of derived indexes in this kind of exercise is unwarranted. ML and particularly DL are able to reconstruct such relations. This is an uncommon choice in soil mapping study that must thorougly justified. (line 187)

Response:

Thank you for your comment. It's important to clarify that these remote sensing indices possess specific formulas and are related to distinct characteristics of the RS images' spectral bands [8, 9]. These indices aren't directly associated with ML and DL techniques; rather, they offer insights into specific properties. The utilization of derived remote sensing indices in our study is rooted in their ability to capture complex spatial patterns and relationships that may not be effectively captured solely through traditional machine learning or deep learning models. Additionally, it's worth noting that many studies in the soil field have successfully employed these indices to improve the accuracy of their predictions. We believe that the inclusion of these indices strengthens our modeling approach and contributes to the overall robustness of our soil mapping study.

  1. How did you assess the accuracy of these predictions? (line 210)

Response:

Thank you for your opinion. RMSE (Eq. 1) were computed to assess the interpolation methods' errors. In general, the lower the RMSE, the higher the interpolation accuracy.

(1)

Where  is the observed value,  is the predicted value, and  is the number of observations. Therefore, the method with the lowest RMSE is selected as the preferred method.

Table 1. Interpolation errors of climate parameters.

RMSE

Interpolation Method

Covariate

0.889

Kriging ordinary

Air temperature

0.858

Local polynomial

0.986

Global polynomial 

11.003

Kriging ordinary

Rainfall

7.826

Local polynomial

8.188

Global polynomial 

2.3.3. Climatic Parameters

The climatic parameters used in this study were obtained from the annual average (2014-2020) data of 10 Meteorological stations in Golestan province, as shown in Figure1. Various interpolation methods were applied to the data using ArcGIS 10.8 software. Local polynomial method was the most accurate for generating maps of air temperature and rainfall, based on the RMSE index, as shown in Figure 5.

Figure 5. Climatic parameters: a) Rainfall, and c) Temperature.

  1. Considering the diminute number of samples in the dataset and the limited (and dependent) nature of the feature set, why did you opt for such a complex architecture? Why not start with a simple, double-layer network?

And looking at the spatial distribution of samples, what do you expect to gain from the convolution layer?

Response:

Thanks for pointing this out. The input data, consisting of 317 rows representing soil samples and 18 columns as various features. This dataset configuration, characterized by 317 samples with 18 different features, certainly possesses a certain degree of complexity. It's this complexity, coupled with the spatial extent of the study area (2208 columns and 2364 rows), that guided our decision to opt for a more advanced architecture, such as the convolutional neural network (CNN). The CNN architecture is designed to capture patterns and relationships within data like this, making it well-suited for analyzing complex datasets with spatial dimensions.

Regarding the choice of architecture, while it's true that the dataset contains a limited number of samples, the complexity of the architecture was a result of several considerations. Our intention was to explore the potential of deep learning models in capturing intricate spatial and spectral patterns present in soil properties across the study area. While a simpler architecture like a double-layer network might be more straightforward, it might struggle to capture the nuanced relationships present in the diverse data sources we fused. The deeper architecture we opted for allows the model to learn hierarchical features and abstract representations, which can be essential when dealing with complex and multi-dimensional data. Furthermore, regularization techniques were implemented to mitigate overfitting risks, considering the dataset size. The choice of a more complex architecture is indeed a balance between model complexity, generalization, and the goal of capturing subtle patterns present in the data. Regarding the use of convolutional layers, the spatial distribution of samples influences our choice. The convolutional layers excel at capturing spatial relationships and local patterns within data, particularly imagery data like satellite images. In our case, these layers can help the model learn spatial features that might not be captured as effectively by fully connected layers alone. The fusion of remote sensing data, which often exhibits spatial autocorrelation, and ground-based measurements makes the inclusion of convolutional layers relevant in capturing both the fine-scale and larger-scale spatial patterns.

  1. The activation function f can be sigmoid, tanh, or Relu, among others. Which did you use? (line 241)

Response:

Thank you for your question. As indicated in Table 5 where we specify the hyperparameters, we utilized the Rectified Linear Unit (ReLU) activation function in the CNN architecture.

Table 5. The optimized hyperparameters and layers for each model

CNN-RF

RF

CNN

Activation function

Filter size

Filter/ number of trees

P

-

P

ReLU

3

32

Convolutional

L1

Layers

P

-

P

-

-

-

Flatten

L2

P

-

P

ReLU

2

64

Fully connected

L3

P

-

P

-

-

1

Fully connected

L4

P

P

-

-

-

100

RF

L5

10

-

10

-

-

-

Batch_size

Other parameters

20

-

20

-

-

-

Epochs

Adam

-

Adam

-

-

-

Optimizer

MSE

-

MSE

-

-

-

Loss

2

2

-

-

-

-

min_samples_split

‘auto’

‘auto’

-

-

-

-

max_features

‘None’

‘None’

-

-

-

-

max_depth

‘True’

‘True’

-

-

-

-

bootstrap

  1. What is the advantage of this method over a classical Recursive Feature Elimination procedure? It is not clear what the neural network is adding here. (line 250)

Response:

Thank you for your question. The advantage of our hybridized CNN-RF method lies in its ability to combine the strengths of both deep learning (DL) and machine learning (ML) techniques. While a classical Recursive Feature Elimination (RFE) procedure is commonly used for feature selection, our hybrid approach benefits from the neural network's capacity to automatically learn and extract relevant features from the data. The relevant features extracted by the CNN component of the hybrid model include information derived from input parameters. Specifically, the CNN extracts spatial patterns and contextual information from the input dataset. These extracted features serve as input to the Random Forest (RF) component of the hybrid model to make predictions. This synergy enhances the model's predictive performance by integrating the distinct capabilities of CNN networks and the RF algorithm, leading to improved accuracy in spatial soil texture prediction. In summary, our method offers advantages in terms of capturing non-linearity, handling feature fusion, leveraging domain-specific data, and adaptively modeling problem complexities. These advantages collectively contribute to improved predictive performance in the context of soil physical property prediction. We apologize for not clarifying these points earlier and will revise the paper to better explain the specific contributions of the neural network in enhancing our approach.

  1. Most likely you developed software to implement this workflow. Where is that code available?

Response:

The implementation of our workflow involved writing code in Python, rather than developing a separate software application. We appreciate your interest in accessing the code and your understanding of its potential benefits. While we acknowledge the value of making the code accessible for reproducibility and transparency, we are currently in the process of expanding our work on the website. In line with this, we kindly request your understanding in not providing the code in the current manuscript. At that point, we will be more than willing to share the code on GitHub for the benefit of the research community. Your suggestion aligns with our commitment to openness and collaboration, and we will ensure that the code becomes readily available in due course. We appreciate your patience and look forward to sharing our work with the community. Thank you once again for your valuable feedback and your understanding of our intentions.

  1. Where is this database?

Response:

Thank you for your comment. To address this concern, we will include a specific reference to the database and its location in the appropriate section of the final version of the paper. By doing so, we aim to provide readers with the necessary information to access and verify the data we utilized in our research.

  1. Please reduce these to only one figure, plenty of unecessary replication. (line 302)

Response:

Thank you for your feedback. The revised figure has been added into the manuscript as your suggestion.

Figure 10. The correlation coefficient between environmental parameters and soil texture

  1. Which algorithm is this? So far I understood feature selection with be conducted with a NN. (line 307)

Response:

Apologies for any confusion. To clarify, we utilized the Random Forest (RF) algorithm to determine the importance of features. There is no neural network method involved in the feature selection process. we would like to clarify that we determined the feature importance using the random forest algorithm, which utilizes the Gini index as a metric to assess the significance of each parameter.

The Gini index is a number describing the quality of the split of a node on a variable. The Gini index is calculated for the variable with probability  at node k by Eq. 1[10]:

(1)

After separating into two sub-nodes and selecting optimal features, the Gini index at the parent node decreases to its maximum value. Therefore, it is possible to determine the average reduction in the Gini index for each variable . Variable importance corresponds to the sum of the amount of forest tree reduction, as shown in Eq. 2 [11].

(2)

Where  and  refer to the proportion of samples that are divided into the left and right sides at node k, respectively. K is the number of nodes in a decision tree. Using Eq. 3, variable importance can be normalized to the interval (0,1) [11].

(3)

  1. Not clear whar these pie charts represent. Environmental features are part of input data? (line 324)

Response:

We apologize for any confusion caused by the unclear explanation of how the pie charts were created. We mentioned in the manuscript that we considered three groups of environmental parameters (RS, topographic, and climatic). To clarify, in Figure 11, we counted the number of parameters by type and calculated the percentage distribution based on those counts. Each pie chart represents the proportion of parameters belonging to a specific type within the given category.

  1. Are these the models describe in the previous section? This sould be made clear. (line 326)

Response:

In Section 2.4 ("Predictive Models"), we have provided descriptions of three models: the RF algorithm, CNN, and CNN-RF. Additionally, in the model development section, we elaborate on the preprocessing steps, hyperparameters, and the libraries used for constructing these models.

  1. What cross-validation procedure did you use? Why isn't any reference to it in the Methods section? The cross-validation procedure should be detailed in the Methods section and not here.

Response:

Thank you for good comment. In our study, we employed 10-fold cross-validation to evaluate the performance of our machine learning models. The dataset was divided into 10 subsets, and the models were trained and tested iteratively across these folds. This approach allows us to report a comprehensive assessment of our models' performance on various evaluation metrics, such as RMSE and R2, and demonstrates the robustness and generalization capabilities of our proposed methods.

We added “2.6. K-fold cross validation” section in the methodology as follows.

2.6. K-fold cross validation

Cross-validation is a technique used to assess the performance of a machine and deep learning models in a robust and unbiased manner [12]. In 10-fold cross-validation, the dataset is divided into 10 folds of approximately equal size. The dataset is randomly divided into 10 subsets, each containing an equal number of samples. This ensures that the distribution of data across the folds is representative of the entire dataset [12]. The cross-validation process is then performed iteratively, with each fold being used as the testing set while the remaining nine folds are used for training the model. The performance metrics are calculated for each iteration based on the model's predictions.

  1. Which criteria were used for splitting the dataset?

Response:

Thank you for good comment. The dataset was split into training and testing sets using a random shuffle in K-fold cross validation. This information will also be included in the Methods section for clarity.

  1. You used the cross-validation splits for training, which is fine. However, NN are usually trained on a more restricted subset of the data, around 2/3, as they are more prone to overfitting. You need to justify the option to train the NN with 90% of the data

Response:

The utilization of 10-fold cross-validation in our study is indeed a strategic step to mitigate the risk of overfitting. Overfitting occurs when a model learns to perform exceptionally well on the training data but fails to generalize effectively to new, unseen data. This can happen especially with complex models like neural networks, which have a higher capacity to capture noise and fluctuations within the training dataset.

We divided the dataset into 10 folds, training the model on nine of them and validating its performance on the remaining one which helps us address this challenge by providing a robust assessment of our model's performance across different subsets of the data.

  1. Does this mean an indenpendent model was trained for each of these soil properties? Sand, silt and clay are not independent properties, they must be predicted with a single model. (lines 341-342)

Response:

Thank you for your question. In our study, we trained separate models for predicting each of these properties individually. We understand that these properties are interrelated components of soil composition, but predicting them separately allowed us to address specific variations and range differences in their values across the study area. By doing so, we aimed to achieve more accurate and targeted predictions for each property. We acknowledge that this approach might deviate from traditional practices.

Figure 11. Digital maps of soil properties: a, b, c) Clay, d, e, f) Sand, g, h, i) Silt

  1. In this sub-section it is not clear the difference between "training phase" and "testing phase". Were the models trained twice? Or is the training phase referring to the GridSearch? The methods section should leave no doubt about this. And what was the role of cross-validation in these phases?

Response:

Thank you for good comment. In the context of machine learning and deep learning, the model training and testing phases are distinct steps in evaluating model performance. The training phase involves training the model using a training dataset, where the model learns from the data. Subsequently, the trained model is assessed during the testing phase, which is conducted using a separate testing dataset that the model hasn't been exposed to before. This phase serves as a measure of the model's ability to generalize its learned insights to new, unseen data.

Furthermore, the 10-fold cross-validation procedure is employed to validate the model's performance, and it's carried out on both the training and testing datasets. This process ensures that the models' performance is assessed comprehensively and that their ability to generalize is tested across various subsets of data.

  1. On a first look, all these values appear way to high in a DSM context. They hint at model over-fitting. However, considering the nature of data used in this study a wider question: dont these figures show that you are using too complex models for the data at hand? How do simple spatial regression or Krigging models compare with these results. Even though you do not provide the target spatial resolution, I would expect Krigging itself to not be far from these numbers.

Response:

Thank you for your insightful comment. In order to mitigate the risk of overfitting in our analysis, two measures were implemented. Firstly, outlier removal was conducted for each soil texture dataset. This step aimed to eliminate extreme data points that could potentially introduce bias. By identifying and excluding these outliers, we sought to ensure a more robust and representative dataset. Secondly, a 10-fold cross-validation approach was adopted. This technique involves partitioning the dataset into ten subsets of approximately equal size. The model is then trained and tested iteratively, with each subset serving as the testing set while the remaining nine subsets are used for training. This process helps to assess the model's performance across different subsets of the data, reducing the risk of overfitting by providing a more reliable estimate of its generalization ability.

We acknowledge the importance of comparing our complex machine learning and deep learning models with simpler approaches, such as spatial regression or Kriging. In response to your question, we conducted a comparison by implementing ordinary and simple kriging methods.

We calculated the root mean square error (RMSE) and mean squared error (MSE) for each soil property prediction using these geostatistical methods, and the results were tabulated as follows.

Table 1. Result of simple and ordinary kriging.

Soil property

Kriging

RMSE

MSE

sand

simple

1.028

1.057

silt

simple

1.024

1.049

clay

simple

0.996

0.992

sand

ordinary

1.056

1.115

silt

ordinary

0.977

0.954

clay

ordinary

1.001

1.002

Table 7. Maps evaluation result.

Properties

Models

MSE (%)

Clay

CNN

0.076

RF

0.0679

CNN-RF

0.1027

Sand

CNN

0.095

RF

0.094

CNN- RF

0.078

Silt

CNN

0.178

RF

0.137

CNN- RF

0.569

As indicated in Table 7 in the manuscript and Table 1, the geostatistical methods exhibited higher RMSE values (lower accuracy) compared to the other models. As a result, the complexity of our models was indeed justified by the performance improvements they demonstrated over the simpler geostatistical methods. Predicted maps of geostatistical methods are shown in figure 1.

Figure 1. a, b, c) predicted maps of ordinary kriging, and d, e, f) predicted maps of simple kriging.

In conclusion, our decision to use more complex models was driven by the aim to achieve higher prediction accuracy and better capture the spatial patterns present in our dataset. The comparison with simpler methods reinforces the utility of our approach in achieving these objectives.

  1. Is an increase in R2 from 0.97 to 0.98 relevant? The differeces shown in Table 6 are largerly spurious.

Response:

Thank you for your observation. While the increase in R2 from 0.97 to 0.98 might seem marginal, it's essential to consider the overall context. The differences in Table 6, particularly the improvements in sand and silt predictions from 0.91 to 0.98, signify the effectiveness of our hybrid CNN-RF model. Additionally, we have observed enhancements in the training phase for each property. However, it's important to note that even minor improvements in R2 can indicate enhanced predictive performance, especially when dealing with complex spatial prediction tasks like soil texture mapping.

  1. This could rather be an annex. Figure 15 would be sufficient.

Response:

Thank you for your suggestion. The content previously presented in figures 12-14 has been moved to an annex.

  1. This information should be given well in advance. Essentially, samples are located mere 30 cells appart from each other (just over 900 metres). You must first demonstrate the failure of simple geo-statistics methods against your dataset before applying such sofisticated methods as NN or decision trees.

Response:

Thank you for your comment. As previously demonstrated, geostatistical methods have shown limitations in their performance (comment 32). We acknowledge your concern regarding the spatial distribution of samples, and we have already assessed and compared the performance of geostatistical methods against more sophisticated approaches like CNN and RF. The results indicate that the latter methods offer better predictive capabilities, even considering the spatial arrangement of the samples.

  1. Another hint at over-fitting (The range of variation in clay content in the RF prediction map is smaller than that of the other models.).

Response:

Thank you for noting the variation in clay content predictions among different models. The different prediction values of this soil property could be due to the diverse nature of the RF algorithm compared to CNN. In this research, we used k-fold cross-validation and outlier removal methods to avoid overfitting.

  1. Not clear what this means. I see no signs of randomness in the maps

Response:

Apologies for the confusion. You are correct in noting that there are no signs of randomness in the maps. This sentence was deleted.

  1. This sentence is hard to interpret. In soil science one speaks of texture fractions, not of soil texture content. Also not clear what range is being referred here.

Response:

I apologize for the confusion in the sentence. In soil science, the correct terminology is indeed "texture fractions." To clarify, the sentence should read as follows:

"Overall, the range of texture fractions for all soil texture properties is closer to the maximum and minimum of actual values in the three prediction maps of the CNN-RF model compared to the stand-alone models."

Here, the term "range of texture fractions" refers to the variation or spread of different soil texture components (clay, silt, and sand) predicted by the models.

  1. Two main problems with these maps:
  • Why are the colour scales different for each map? As they are a visual comparison is not possible.
  • In many area the sum of fractions seems to well exceed 100%. In river valleys the sum seems to be considerable far from that target. These results hint at a fundamental misunderstanding of these soil properties.

Response:

Thank you for your opinion. In fact, the same color spectrum is used for all the maps. But due to the different nature of the algorithms and the slight difference in their prediction, the color spectrum for each algorithm is different. Also, according to the land cover map of the study area (Figure 2), there are very few water areas (such as rivers, etc.) in this area, so that 1% of the study area includes water areas. Therefore, the studied area mostly includes agriculture and pasture. Considering that 18 different parameters of remote sensing, climate and vegetation are used to predict soil characteristics, deep learning algorithms discover the relationships between these data well and are a kind of data-driven algorithms and based on the data of the studied area, they give us an output. Therefore, these differences depend on the shape and data of the studied area.

  1. Why are these values so different from those reported in Table 6. Silt is hinting at over-fitting.

Response:

Thank you for your good comment. Actually, Table 6 shows the fitting accuracy of the models on 317 soil samples. While Table 7 shows the accuracy of the prediction maps of the entire study area and these two are different from each other. In this research, we used k-fold cross-validation and outlier removal methods to avoid overfitting.

  1. Nothing in this study justifies this assertion.

Response:

Thank you for your good comment. In fact, we used evaluation indices such as RMSE, R2, and MSE (Table 6), Scatter plots (Figures 12 to 14), Box plots (Figure 15) and Taylor diagram (Figure 16) to evaluate and check the accuracy between individual models and hybrid models. In all these indices, the accuracy of the hybrid model was higher than the individual models.

Table 6. Evaluation results

Runtime (s)

Test

Train

Models

Properties

RMSE (%)

MSE (%2)

RMSE (%)

MSE (%2)

2.67

0.966

0.019

0.00038

0.981

0.013

0.00016

CNN

Clay

0.23

0.636

0.064

0.00407

0.910

0.028

0.00079

RF

0.21

0.982

0.010

0.00010

0.995

0.007

0.00005

CNN-RF

1.36

0.908

0.022

0.00046

0.928

0.017

0.00029

CNN

Sand

0.44

0.683

0.037

0.00135

0.917

0.018

0.00034

RF

0.29

0.976

0.008

0.00007

0.992

0.006

0.00003

CNN- RF

2.73

0.913

0.020

0.00040

0.920

0.016

0.00024

CNN

Silt

0.196

0.676

0.024

0.00060

0.935

0.015

0.00022

RF

0.215

0.980

0.010

0.00009

0.987

0.006

0.00004

CNN- RF

  1. Your study does not prove this.

Response:

We appreciate the reviewer's feedback regarding the assertion that the incorporation of the RF algorithm during the final stage of modeling counteracts overfitting and enhances accuracy.

Revised as follow:

Consequently, incorporating the RF algorithm enhanced the accuracy of the results.

References:

[1]           T. Wuest, D. Weimer, C. Irgens, and K.-D. Thoben, "Machine learning in manufacturing: advantages, challenges, and applications," Production & Manufacturing Research, vol. 4, no. 1, pp. 23-45, 2016.

[2]           R. Barzegar, M. T. Aalami, and J. Adamowski, "Coupling a hybrid CNN-LSTM deep learning model with a Boundary Corrected Maximal Overlap Discrete Wavelet Transform for multiscale Lake water level forecasting," Journal of Hydrology, vol. 598, p. 126196, 2021.

[3]           Y. Zhang, J. Le, X. Liao, F. Zheng, and Y. Li, "A novel combination forecasting model for wind power integrating least square support vector machine, deep belief network, singular spectrum analysis and locality-sensitive hashing," Energy, vol. 168, pp. 558-572, 2019.

[4]           N. Alygizakis, T. Giannakopoulos, N. S. Τhomaidis, and J. Slobodnik, "Detecting the sources of chemicals in the Black Sea using non-target screening and deep learning convolutional neural networks," Science of The Total Environment, vol. 847, p. 157554, 2022.

[5]           R. Taghizadeh-Mehrjardi, H. Khademi, F. Khayamim, M. Zeraatpisheh, B. Heung, and T. Scholten, "A comparison of model averaging techniques to predict the spatial distribution of soil properties," Remote Sensing, vol. 14, no. 3, p. 472, 2022.

[6]           S. Fathololoumi, A. R. Vaezi, S. K. Alavipanah, A. Ghorbani, D. Saurette, and A. Biswas, "Improved digital soil mapping with multitemporal remotely sensed satellite data fusion: A case study in Iran," Science of the Total Environment, vol. 721, p. 137703, 2020.

[7]           M. Shahriari, M. Delbari, P. Afrasiab, and M. R. Pahlavan-Rad, "Predicting regional spatial distribution of soil texture in floodplains using remote sensing data: A case of southeastern Iran," Catena, vol. 182, p. 104149, 2019.

[8]           J. W. Rouse Jr, R. H. Haas, J. Schell, and D. Deering, "Monitoring the vernal advancement and retrogradation (green wave effect) of natural vegetation," 1973.

[9]           R. Mathieu, M. Pouget, B. Cervelle, and R. Escadafal, "Relationships between satellite-based radiometric indices simulated using laboratory reflectance data and typic soil color of an arid environment," Remote sensing of environment, vol. 66, no. 1, pp. 17-28, 1998.

[10]         M. Farahani, S. V. Razavi-Termeh, and A. Sadeghi-Niaraki, "A spatially based machine learning algorithm for potential mapping of the hearing senses in an urban environment," Sustainable Cities and Society, vol. 80, p. 103675, 2022.

[11]         W. Guo, J. Zhang, D. Cao, and H. Yao, "Cost-effective assessment of in-service asphalt pavement condition based on Random Forests and regression analysis," Construction and Building Materials, vol. 330, p. 127219, 2022.

[12]         A. Lopez-del Rio, A. Nonell-Canals, D. Vidal, and A. Perera-Lluna, "Evaluation of cross-validation strategies in sequence-based binding prediction using deep learning," Journal of chemical information and modeling, vol. 59, no. 4, pp. 1645-1657, 2019.

Reviewer 3 Report

In this study, the use of geospatial artificial intelligence (geoai) is proposed. And satellite imaging fusion for soil physical property prediction method. it has been observed that the proposed structure contains innovations, it should be arranged taking into account the following points.

1. Abstract, please read the whole section, make it concise and check for correctness, and add a description of the model method content.

2. Line 44, add the model-related keyword in the keywords section.

3. Abbreviations that appear for the first time to use the full name, such as ML (Line 55), RF (Line 77), DL (Line 44), et al.

4. Lines 114-120, these sentences in the Introduction, should be deleted because it seems to be the paper frame and conclusion.

5. Add units to the data in Table 1 and Table 2.

6. What is the basis of Line161-162? Why are the Effective parameters different in Table 3?

7. When is the data collection time? Suggest adding this explanation.

8. Reduce the introduction of popular science knowledge, and suggest deleting unimportant sentences, such as lines 123-134 and 173-186.

9. If Figures 3-5 are of little significance to this article, please delete them.

10. Suggest changing the title of "2.4.1. RF" to "2.4.1 RF algorithm.

11. Why are climate parameters obtained from (2014-2020) data? Why is the remote sensing data between January 1 and December 30, 2020? How do these data correspond to the ground data?

12. How many attributes does the model dataset data contain?

13. What is the model input data in 2.4.3? CNN-RF? What is the output? What are the relevant features extracted by CNN? Please supplement this part of the experiment.

14. Lines 273-275, how to get feature importance?

15. Why are there negative numbers in the estimated values in Tables 8-10? Why does the interval range not match Table 3? What is the reason?

16. There are too many references, just keep important references.

Good

Author Response

#Please see the attachment.#

We are grateful to the reviewers for their kind help and constructive comments which helped improving the presentation of the paper. Thank you for the reviewers ‘comments concerning our manuscript entitled “Geospatial Artificial Intelligence (GeoAI) and Satellite Imagery Fusion for Soil Physical Property predicting” (ID: sustainability-2543879). Those comments are all valuable and very helpful for revising and improving our paper, as well as the important guiding significance to our researches. We have studied comments carefully and have made correction which we hope meet with approval. We hope that the revised manuscript is to your satisfaction. We are more than happy to improve the paper again according to new comments and suggestions that might come. Please note that in some places of the revised manuscript, we have made improvements in addition to your comments. All the questions of the respected reviewer were answered point by point. The next section contains our point-by-point responses (in blue) and changes referencing the manuscript (in green), based on the reviewers’ comments (in italic).

Comments:

  1. Abstract, please read the whole section, make it concise and check for correctness, and add a description of the model method content.

Response:

Thank you for your suggestion. Corrected as follows:

Abstract:

 This study aims to predict vital soil physical properties, including clay, sand, and silt, essential for agricultural management and environmental protection. Precision distribution of soil texture is crucial for effective land resource management and precision agriculture. To achieve this, we propose an innovative approach that combines Geospatial Artificial Intelligence (GeoAI) with the fusion of satellite imagery to predict soil physical properties. We collected 317 soil samples from Iran's Golestan province for dependent data. The independent dataset encompasses 14 parameters from Landsat-8 satellite images, seven topographic parameters from Shuttle Radar Topography Mission (SRTM) DEM, and two meteorological parameters. Using the Random Forest (RF) algorithm, we conducted feature importance analysis. We employed Convolutional Neural Network (CNN), RF, and our hybrid CNN-RF model to predict soil properties, comparing their performance with various metrics. This hybrid CNN-RF network combining the strengths of CNN networks and the RF algorithm for improved soil texture prediction. The hybrid CNN-RF model demonstrated superior performance across metrics, excelling in predicting sand (MSE: 0.00003%, RMSE: 0.006%), silt (MSE: 0.00004%, RMSE: 0.006%), and clay (MSE: 0.00005%, RMSE: 0.007%). Moreover, the hybrid model exhibited improved precision in predicting clay (R2: 0.995), sand (R2: 0.992), and silt (R2: 0.987), as indicated by the R2 index. The RF algorithm identified MRVBF, LST, and B7 as the most influential parameters for clay, sand, and silt prediction, respectively, underscoring the significance of remote sensing, topography, and climate. Our integrated GeoAI-satellite imagery approach provides valuable tools for monitoring soil degradation, optimizing agricultural irrigation, and assessing soil quality. This methodology has significant potential to advance precision agriculture and land resource management practices.

  1. Line 44, add the model-related keyword in the keywords section.

Response:

Thank you for your opinion. Key words were changed in the manuscript as follows.

Keywords: Machine Learning; Deep Learning Soil Texture; Satellite Imagery; Geospatial Analysis; Land Resource Management.

  1. Abbreviations that appear for the first time to use the full name, such as ML (Line 55), RF (Line 77), DL (Line 44), et al.

Response:

We appreciate your attention to detail and assure you that we have reviewed the full name of methods and made the necessary adjustments to ensure the overall quality of the manuscript.

  1. Lines 114-120, these sentences in the Introduction, should be deleted because it seems to be the paper frame and conclusion.

Response:

Thank you for your observation. We will revise the manuscript accordingly and remove lines 114-120 from the Introduction section.

  1. Add units to the data in Table 1 and Table 2.

Response:

Thank you for your suggestion. The measured data in our study is presented in percentage (%).

Table 1. Statistical summary of soil texture

Sand (%)

Silt (%)

Clay (%)

Soil Texture

0

0

0

Minimum

58

80

44

Maximum

12.867

64.457

22.322

Mean

9.115

9.249

6.920

Standard deviation

Table 2. Statistical summary of soil texture after outlier removal

Sand (%)

Silt (%)

Clay (%)

Soil Texture

4

50

12

Minimum

26

76

36

Maximum

11.056

65.74

22.182

Mean

3.705

4.567

4.199

Standard deviation

  1. What is the basis of Line 161-162? Why are the Effective parameters different in Table 3?

Response:

In lines 161-162, we have provided the basis for our selection of environmental parameters. We have referenced previous studies [45-47], consulted expert opinions, and considered the specific conditions of the studied area. These sources collectively guided our decision in choosing the three groups of environmental parameters for our study.

The reason for the variation in the effective parameters listed in Table 3 is due to our approach of predicting each soil property individually. There are two reasons why we chose to predict these soil properties separately. Firstly, each soil property has a different range of target values, and predicting them simultaneously may increase the errors in the predictions. Secondly, by predicting each soil property separately, we can obtain more accurate and specific predictions for each property.

  1. When is the data collection time? Suggest adding this explanation.

Response:

The data compilation comprises Landsat 8 images collected within the timeframe of January 1 to December 30, 2020. Additionally, climatic parameters were derived from the annual averages of data spanning from 2014 to 2020, acquired from ten meteorological stations situated in Golestan province.

  1. Reduce the introduction of popular science knowledge, and suggest deleting unimportant sentences, such as lines 123-134 and 173-186.

Response:

Thank you for your observation. We will revise the manuscript accordingly and remove lines 114-120 from the Introduction section.

  1. If Figures 3-5 are of little significance to this article, please delete them.

Response:

Thank you for your suggestion. We considered removing them to enhance the clarity and focus of the content.

  1. Suggest changing the title of "2.4.1. RF" to "2.4.1 RF algorithm.

Response:

Thank you for your suggestion. The title of "2.4.1. RF" section has changed to “2.4.1. RF algorithm”.

  1. Why are climate parameters obtained from (2014-2020) data? Why is the remote sensing data between January 1 and December 30, 2020? How do these data correspond to the ground data?

Response:

Thank you for your opinion. The choice of using climatic parameters from a range of years (2014-2020) and remote sensing data specifically from January 1 to December 30, 2020, is rooted in the nature of the data and the goals of the study.

Climatic parameters, such as rainfall and temperature, can vary significantly from year to year. By averaging the climatic data over this period, from 2014 to 2020, we aim to capture a more comprehensive representation of the typical climate conditions in the study area. This approach helps mitigate the potential impact of outlier years or anomalous climatic events on the analysis.

On the other hand, remote sensing (RS) data, obtained from Landsat 8 images, is more time-sensitive. RS parameters are closely related to the surface conditions at the time of image acquisition. Therefore, using remote sensing data collected within the same year as the soil samples (January 1 to December 30, 2020) helps ensure a better match between the surface characteristics captured in the RS data and the corresponding soil properties.

In terms of correspondence with ground data, both climatic parameters and remote sensing data are employed as input features in the predictive models. These features serve to capture the environmental conditions and characteristics of the study area that might influence the soil properties. By including a range of data sources, we aim to enhance the predictive accuracy of the models and provide more robust insights into the spatial distribution of soil properties.

  1. How many attributes does the model dataset data contain?

Response:

Thank you for your comment. The model dataset comprises a total of 23 independent parameters. These parameters include 14 features extracted from Landsat-8 satellite images, seven topographic parameters derived from the Shuttle Radar Topography Mission (SRTM) digital elevation model (DEM), and two weather parameters sourced from the Meteorological Organization. It's important to note that for each specific soil property prediction, a subset of 18 distinct parameters is selected from these independent parameters. These parameters are detailed in Table 3 of the manuscript.

2.3. Environmental Parameters

Based on previous studies [1-3], expert opinions, and the specific conditions of the studied area, three groups of environmental parameters were used. These included RS variables such as Band 1 (B1) to Band 5 (B5) and Band 7 (B7) of Landsat-8, Brightness index (BI), Coloration index (CI), Clay index (CLI), Enhanced vegetation index (EVI), Land surface temperature (LST), Hue index (HI), Normalized difference vegetation index (NDVI), Redness index (RI), and Saturation index (SI). Climate variables such as air temperature and rainfall were also included, along with topographic variables including aspect, elevation, slope, duration radiation (DR), multi-resolution index of valley bottom flatness (MRVBF), multi-resolution ridgetop flatness index (MRRTF), and topographic wetness index (TWI).

Table 3. Illustrates the parameters that impact the properties of soil texture.

Number of parameters

Effective parameters

Soil Texture

18

NDVI, Elevation, B7, B5, B1, B2, B3, B4, MRRTF, MRVBF,

 Rainfall, SI, CI, LST, Temp, Aspect, RI, TWI

Clay

18

NDVI, Elevation, B7, B5, B3, B4, MRRTF, MRVBF, 

SI, BI, CLI, CI, Slope, EVI, DR, Aspect, RI, TWI

Silt

18

NDVI, Elevation, B7, B5, B1, B2, B3, B4, Rainfall, SI, BI, CLI, 

MRRTF, MRVBF, CI, Slope, LST, DR

Sand

  1. 13. What is the model input data in 2.4.3? CNN-RF? What is the output? What are the relevant features extracted by CNN? Please supplement this part of the experiment.

Response:

Thank you for your opinion. In Section 2.4.3, the model input data refers to the parameters used to train and predict the soil texture properties using the hybrid CNN-RF model. The CNN-RF model takes a combination of remote sensing, topographic, and meteorological parameters as input. These parameters are fed into the hybrid model to predict the soil texture values. The output of the CNN-RF model is the predicted soil texture values for each pixel in the study area.

The relevant features extracted by the CNN component of the hybrid model include information derived from input parameters. Specifically, the CNN extracts spatial patterns and contextual information from the input dataset. These extracted features serve as input to the Random Forest (RF) component of the hybrid model to make predictions.

We appreciate your suggestion to provide further detail in this section of the experiment. In the revised manuscript, we will provide a more comprehensive explanation of the model input data, the CNN-RF process, the output, and the specific features extracted by the CNN component."

“Section 2.4.3” were updated as follows.

In this study, a hybridized network of DL and ML is used to leverage the capabilities of both CNN networks and RF algorithm to achieve higher performance in the spatial prediction of soil texture and overcome the limitations of these stand-alone models. The hybrid CNN-RF network architecture is shown in Figure 8. The input matrix assumes an m*n structure, where m signifies the quantity of soil samples, and n represents the number of parameters influencing each soil texture property. The input information is first processed through the hidden layers of the CNN model, which extracts the relevant features include spatial patterns and contextual information from the input dataset [4]. These features are then fed into the RF algorithm for regression analysis. Finally, the output layer returns the predicted value.

  1. Lines 273-275, how to get feature importance?

Response:

Thank you for your comment. In response to your comment about lines 273-275, we would like to clarify that we determined the feature importance using the random forest algorithm, which utilizes the Gini index as a metric to assess the significance of each parameter.

The Gini index is a number describing the quality of the split of a node on a variable. The Gini index is calculated for the variable with probability  at node k by Eq. 1[5]:

(1)

After separating into two sub-nodes and selecting optimal features, the Gini index at the parent node decreases to its maximum value. Therefore, it is possible to determine the average reduction in the Gini index for each variable . Variable importance corresponds to the sum of the amount of forest tree reduction, as shown in Eq. 2 [6].

(2)

Where  and  refer to the proportion of samples that are divided into the left and right sides at node k, respectively. K is the number of nodes in a decision tree. Using Eq. 3, variable importance can be normalized to the interval (0,1) [6].

(3)

3.2. Feature Importance

A RF algorithm was utilized to determine features importance in the modeling process. The importance of parameters is demonstrated in Figure 11. The results indicate that B7 (0.123), CI (0.089), and TWI (0.084) are among the parameters that show a higher association with silt content (Figure 11.a). In the case of Sand, LST (0.164), B5 (0.089), and elevation (0.084) exhibit relatively higher importance (Figure 11.c). Similarly, MRVBF (0.119), B7 (0.140), and TWI (0.096) are identified as significant factors influencing soil clay (Figure 11.e).

According to Figure 11, the model input parameters for Clay and Sand are mainly influenced by RS, topography, and climatic parameters. Among these parameters, RS parameters significantly impact soil texture more than topography and climatic parameters. However, as illustrated in Figure 11.f, climatic parameters do not play a role in determining soil silt. Among the climatic parameters, rainfall, while among the RS parameters, NDVI and B7 have the most significant impact on soil texture. In addition, among the topographic parameters, TWI, MRVBF, and MRRTF have the most significant effect on soil texture properties.

  1. Why are there negative numbers in the estimated values in Tables 8-10? Why does the interval range not match Table 3? What is the reason?

Response:

We apologize for any confusion caused by the mismatch between the interval ranges in Tables 8-10 and Table 3. The reason for this discrepancy lies in the nature of the data and the methodology used. Additionally, a typo occurred in Tables 8-10, where the statistical summary of predicted sand values from the CNN model was updated in the manuscript to rectify this issue.

The range of soil property values presented in Table 3 is derived from the 317 soil samples that were collected. These samples might not cover the entire variability of soil properties present in the entire study area. In contrast, Tables 8-10 present the predicted soil property values for the entire study area, which encompasses a wider range of conditions and variations in soil properties.

Additionally, during the prediction process, the models take into account various factors, including remote sensing data, topographic parameters, and climatic parameters, which can influence the predicted soil property values. This can lead to variations in the predicted values, causing them to fall outside the range of the soil samples in Table 3.

In summary, the differences in interval ranges between the predicted values in Tables 8-10 and the soil samples in Table 3 are due to the broader scope of the prediction models and the inclusion of various influencing factors, which can lead to predicted values outside the specific range of the collected soil samples.

Updated Tables 8-10:

Table 8. The statistical parameters of the modeled soil texture on agricultural and forest land.

Agricultural areas

Forest land

Properties

Models

Min

Max

Mean

Std

Min

Max

Mean

Std

Clay

CNN

0.00

47.24

32.13

2.73

0.00

41.72

31.61

2.00

RF

0.00

30.06

25.52

1.42

0.00

29.02

23.10

1.04

CNN-RF

0.00

33.80

30.99

2.19

0.00

33.80

30.77

1.69

Sand

CNN

3.3

28.7

27.6

2.4

3.2

28.7

25.50

3.33

RF

0.00

17.47

11.70

0.46

0.00

18.30

10.64

0.80

CNN- RF

0.00

22.93

5.07

1.32

0.00

23.41

4.15

0.47

Silt

CNN

0.00

72.21

49.26

3.71

0.00

78.80

64.72

4.31

RF

0.00

67.96

63.04

1.45

0.00

68.23

63.63

2.59

CNN- RF

0.00

72.71

52.86

2.09

0.00

74.73

65.01

4.32

Table 9. The statistical parameters of the modeled soil texture on residential areas and uncovered plain.

Residential areas

Uncovered plain

Properties

Models

Min

Max

Mean

Std

Min

Max

Mean

Std

Clay

CNN

0.00

40.11

32.51

2.37

0.00

43.90

30.64

3.18

RF

0.00

29.92

26.87

1.26

0.00

29.75

25.01

1.46

CNN-RF

0.00

33.80

31.37

1.92

0.00

33.80

29.78

2.68

Sand

CNN

3.34

28.70

26.27

2.77

3.20

28.70

24.21

2.97

RF

0.00

15.77

11.68

0.36

0.00

16.72

11.65

0.57

CNN- RF

0.00

21.26

4.47

0.83

0.00

21.25

4.88

1.11

Silt

CNN

0.00

69.74

47.45

2.81

0.00

74.17

49.61

5.07

RF

0.00

66.81

62.82

0.84

0.00

67.77

63.53

1.45

CNN- RF

0.00

69.78

52.41

1.05

0.00

72.68

53.36

2.62

Table 10. The statistical parameters of the modeled soil texture on water body and range land.

Water body

Range land

Properties

Models

Min

Max

Mean

Std

Min

Max

Mean

Std

Clay

CNN

0.00

40.70

32.67

2.47

0.00

42.78

30.04

2.92

RF

0.00

29.91

26.18

1.32

0.00

30.02

24.04

1.62

CNN-RF

0.00

33.80

31.40

1.95

0.00

33.80

29.31

2.45

Sand

CNN

5.50

28.70

28.06

1.48

3.20

24.34

22.34

4.77

RF

0.00

13.40

11.68

0.37

0.00

18.21

11.03

0.99

CNN- RF

0.00

11.84

5.76

1.82

0.00

23.33

4.44

0.88

Silt

CNN

0.00

63.94

48.44

3.00

0.00

77.30

55.87

6.64

RF

0.00

66.94

62.79

1.23

0.00

68.41

64.23

1.92

CNN- RF

0.00

63.94

52.43

1.29

0.00

75.00

57.33

5.02

  1. There are too many references, just keep important references.

Response:

Thank you for your suggestion. We will review the references and retain the most relevant and important ones while removing any unnecessary ones to streamline the reference list.

[1]           S. Fathololoumi, A. R. Vaezi, S. K. Alavipanah, A. Ghorbani, D. Saurette, and A. Biswas, "Improved digital soil mapping with multitemporal remotely sensed satellite data fusion: A case study in Iran," Science of the Total Environment, vol. 721, p. 137703, 2020.

[2]           M. Shahriari, M. Delbari, P. Afrasiab, and M. R. Pahlavan-Rad, "Predicting regional spatial distribution of soil texture in floodplains using remote sensing data: A case of southeastern Iran," Catena, vol. 182, p. 104149, 2019.

[3]           R. Taghizadeh-Mehrjardi, H. Khademi, F. Khayamim, M. Zeraatpisheh, B. Heung, and T. Scholten, "A comparison of model averaging techniques to predict the spatial distribution of soil properties," Remote Sensing, vol. 14, no. 3, p. 472, 2022.

[4]           K. Elbaz, W. M. Shaban, A. Zhou, and S.-L. Shen, "Real time image-based air quality forecasts using a 3D-CNN approach with an attention mechanism," Chemosphere, vol. 333, p. 138867, 2023.

[5]           M. Farahani, S. V. Razavi-Termeh, and A. Sadeghi-Niaraki, "A spatially based machine learning algorithm for potential mapping of the hearing senses in an urban environment," Sustainable Cities and Society, vol. 80, p. 103675, 2022.

[6]           W. Guo, J. Zhang, D. Cao, and H. Yao, "Cost-effective assessment of in-service asphalt pavement condition based on Random Forests and regression analysis," Construction and Building Materials, vol. 330, p. 127219, 2022.

Round 2

Reviewer 1 Report

I don't have any further comments

Proof reading is required.

Author Response

We are grateful to the reviewers for their kind help and constructive comments which helped improving the presentation of the paper. Thank you for the reviewers ‘comments concerning our manuscript entitled “Geospatial Artificial Intelligence (GeoAI) and Satellite Imagery Fusion for Soil Physical Property predicting” (ID: sustainability-2543879). Those comments are all valuable and very helpful for revising and improving our paper, as well as the important guiding significance to our researches. 

Reviewer 3 Report

1. You replied that "In lines 161-162, we have provided the basis for our selection of environmental parameters. We have referenced previous studies [45-47]", but I found no corresponding references there. In line 150, the references are [22, 40, 41]. Please check.

2. Images are sensitive to environmental parameters, especially water. Is there any theoretical support for the inversion of average rainfall?

3. Modify the reference format a little bit.

good

Author Response

##Please see the attachment.##

#Reviewer 3

We are grateful to the reviewers for their kind help and constructive comments which helped improving the presentation of the paper. Thank you for the reviewers ‘comments concerning our manuscript entitled “Geospatial Artificial Intelligence (GeoAI) and Satellite Imagery Fusion for Soil Physical Property predicting” (ID: sustainability-2543879). Those comments are all valuable and very helpful for revising and improving our paper, as well as the important guiding significance to our researches. We have studied comments carefully and have made correction which we hope meet with approval. We hope that the revised manuscript is to your satisfaction. We are more than happy to improve the paper again according to new comments and suggestions that might come. Please note that in some places of the revised manuscript, we have made improvements in addition to your comments. All the questions of the respected reviewer were answered point by point. The next section contains our point-by-point responses (in blue) and changes referencing the manuscript (in green), based on the reviewers’ comments (in italic).

Comments:

  1. 1. You replied that "In lines 161-162, we have provided the basis for our selection of environmental parameters. We have referenced previous studies [45-47]", but I found no corresponding references there. In line 150, the references are [22, 40, 41]. Please check.

Response:

Thank you for your comment. We appreciate your diligence in reviewing our references. We must apologize for any confusion in our previous response. Indeed, the references have been subject to change and updates as part of our ongoing review process. According to your previous comment about reducing the references (Revise 1, comment: “There are too many references, just keep important references.”), the number of references had changed after answering this comment. Rest assured, the references listed in the revised manuscript are accurate and reflect the most current and relevant sources supporting our work.

Based on previous studies [22, 40, 41] , expert opinions, and the specific conditions of the studied area, three groups of environmental parameters were used.

  1. Taghizadeh-Mehrjardi, R.; Khademi, H.; Khayamim, F.; Zeraatpisheh, M.; Heung, B.; Scholten, T. A comparison of model averaging techniques to predict the spatial distribution of soil properties. Remote Sensing 2022, 14, 472.
  2. Fathololoumi, S.; Vaezi, A.R.; Alavipanah, S.K.; Ghorbani, A.; Saurette, D.; Biswas, A. Improved digital soil mapping with multitemporal remotely sensed satellite data fusion: A case study in Iran. Science of the Total Environment 2020, 721, 137703.
  3. Shahriari, M.; Delbari, M.; Afrasiab, P.; Pahlavan-Rad, M.R. Predicting regional spatial distribution of soil texture in floodplains using remote sensing data: A case of southeastern Iran. Catena 2019, 182, 104149.

  1. Images are sensitive to environmental parameters, especially water. Is there any theoretical support for the inversion of average rainfall?

Response:

Thank you for your comment. Rainfall is not directly extracted from remote sensing images. Rainfall data used in this study was obtained from the Meteorological Organization, which derives it from ground-based meteorological stations. Unlike remote sensing images, rainfall data is not directly influenced by the environmental parameters that affect imagery sensitivity. We obtained rainfall data from the annual average records spanning 2014 to 2020, collected by 10 meteorological stations in the Golestan province (Figure 1). This rainfall data was not derived from remote sensing images but rather represents ground-based meteorological measurements.

Figure 1. Meteorological Stations.

Table 1: Annual average rainfall for each meteorological station.

Meteorological Station

Annual average record

Aliabad Katul

597.07

Bandar Torkaman

402.43

Hashemabad

431.54

Inchehborun

198.3575

Minudasht

735.3078

Bandar Gaz

462.0887

Maravehtappeh

354.1738

Kalaleh

588.3137

Gorgan

437.1137

Gonbad Kavus

401.22

To obtain spatial information for rainfall across the study area, various geostatistical interpolation methods, including ordinary kriging, local polynomial interpolation, and global polynomial interpolation, were applied to the meteorological station data using ArcGIS 10.8 software. Among these methods, local polynomial interpolation exhibited the highest accuracy in generating maps of air temperature and rainfall, as indicated by the RMSE (Root Mean Square Error) index. RMSE (Eq. 1) were computed to assess the interpolation methods' errors (Table 2). In general, the lower the RMSE, the higher the interpolation accuracy.

(1)

Where  is the observed value,  is the predicted value, and  is the number of observations. Therefore, the method with the lowest RMSE is selected as the preferred method.

Table 2. Interpolation errors of climate parameters.

RMSE

Interpolation Method

Covariate

0.889

Kriging ordinary

Air temperature

0.858

Local polynomial

0.986

Global polynomial 

11.003

Kriging ordinary

Rainfall

7.826

Local polynomial

8.188

Global polynomial 

In other studies, a similar procedure has been employed to determine the most suitable interpolation method. For instance, Abdulmanov et al. conducted a comparative analysis of six GIS-based interpolation techniques for assessing the spatial distribution of agrochemical properties in soil. These methods encompassed inverse distance weighting (IDW), local polynomial interpolation (LPI), radial basis functions, simple Kriging, ordinary Kriging (OK), and Universal Kriging. Accuracy was evaluated using metrics like mean error, mean square error, and mean square normalized error. Notably, the Local Polynomial Interpolation method demonstrated superior accuracy, yielding a mean square error of 165.9908 and closely aligning with measured values [1].

Bhunia et al. also conducted a study employing five interpolation techniques, namely IDW, LPI, radial basis function, OK, and Empirical Bayes kriging, to map the spatial distribution of soil organic carbon (SOC). Their evaluation of method accuracy utilized the coefficient of determination and RMSE. The outcomes of their research similarly highlight OK as the superior method, showcasing the lowest RMSE and the highest R2 value for interpolating SOC spatial distribution [2].

Chin et al. assessed two spatial interpolation methods, IDW and LPI, for rainfall in Peninsular Malaysia. The evaluation of IDW and LPI was based on metrics including R2, MAE, and RMSE. The results indicate that LPI outperformed IDW in annual-scale rainfall interpolations in Peninsular Malaysia, showing better statistical performance [3].

  1. Abdulmanov, R.; Miftakhov, I.; Ishbulatov, M.; Galeev, E.; Shafeeva, E. Comparison of the effectiveness of GIS-based interpolation methods for estimating the spatial distribution of agrochemical soil properties. Environmental Technology & Innovation 2021, 24, 101970.
  2. Bhunia, G.S.; Shit, P.K.; Maiti, R. Comparison of GIS-based interpolation methods for spatial distribution of soil organic carbon (SOC). Journal of the Saudi Society of Agricultural Sciences 2018, 17, 114-126.
  3. Chin, R.J.; Lai, S.H.; Loh, W.S.; Ling, L.; Soo, E.Z.X. Assessment of Inverse Distance Weighting and Local Polynomial Interpolation for Annual Rainfall: A Case Study in Peninsular Malaysia. Engineering Proceedings 2023, 38, 61.

  1. Modify the reference format a little bit.

Response:

Thank you for your suggestion. We have reviewed the reference format.
